# THE ILLUSION OF DIMINISHING RETURNS: MEASURING LONG HORIZON EXECUTION IN LLMS

**Akshit Sinha**[1*]     **Arvindh Arun**[2*]     **Shashwat Goel**[3,4*]
**Steffen Staab**[2,5]     **Jonas Geiping**[3,4,6]

[1]University of Cambridge     [2]Institute for AI, University of Stuttgart
[3]Max Planck Institute for Intelligent Systems     [4]ELLIS Institute Tübingen
[5]University of Southampton [6]Tübingen AI Center
[*]Equal contribution

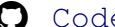 Code          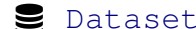 Dataset

## ABSTRACT

Does continued scaling of large language models (LLMs) yield diminishing returns? In this work, we show that short-task benchmarks may give an illusion of slowing progress, as even marginal gains in single-step accuracy can compound into exponential improvements in the length of tasks a model can successfully complete. Then, we argue that failures of LLMs when simple tasks are made longer arise from mistakes in *execution*, rather than an inability to *reason*. So, we propose isolating *execution* capability, by explicitly providing the *knowledge* and *plan* needed to solve a long-horizon task. First, we find that larger models can correctly execute significantly more turns even when small models have near-perfect single-turn accuracy. We then observe that the per-step accuracy of models degrades as the number of steps increases. This is not just due to long-context limitations—curiously, we observe a *self-conditioning* effect—models become more likely to make mistakes when the context contains their errors from prior turns. Self-conditioning does not reduce by just scaling the model size. But, we find that *thinking* mitigates self-conditioning, and also enables execution of much longer tasks in a single turn. We conclude by benchmarking frontier thinking models on the length of tasks they can execute in a single turn. Overall, by focusing on the ability to execute, we hope to reconcile debates on how LLMs can solve complex reasoning problems yet fail at simple tasks when made longer, and highlight the massive benefits of scaling model size and sequential test-time compute for long-horizon tasks.

## 1 INTRODUCTION

Is continued scaling of compute for Large Language Models (LLMs) economically justified given diminishing marginal gains? This question lies at the heart of the ongoing debate on the viability of continued massive investments in LLMs. While scaling laws show diminishing returns on metrics like test loss, the true economic potential of LLMs might arise from automating long, multi-step tasks (METR, 2025). However, long-horizon tasks have been the Achilles' heel of Deep Learning. We see recent vision models generating impressive images, yet consistency over long videos remains an unsolved challenge. As the industry races to build agents that tackle entire projects, not just isolated questions, a fundamental question arises:

*How can we measure the number of steps an LLM can reliably execute?*

LLM failures on simple, but long tasks have been considered a fundamental inability to *reason* (Mirzadeh et al., 2024). Despite massive improvements on complex reasoning benchmarks, Shojaee et al. (2025) claim *thinking models* (Guo et al., 2025) only give an "illusion of thinking", as they eventually fail when the task is made longer. These results have sparked much debate in the community, which we think can be resolved by decoupling the need for *planning* and *execution* in reasoning or agentic tasks. *Planning* involves deciding what information to retrieve

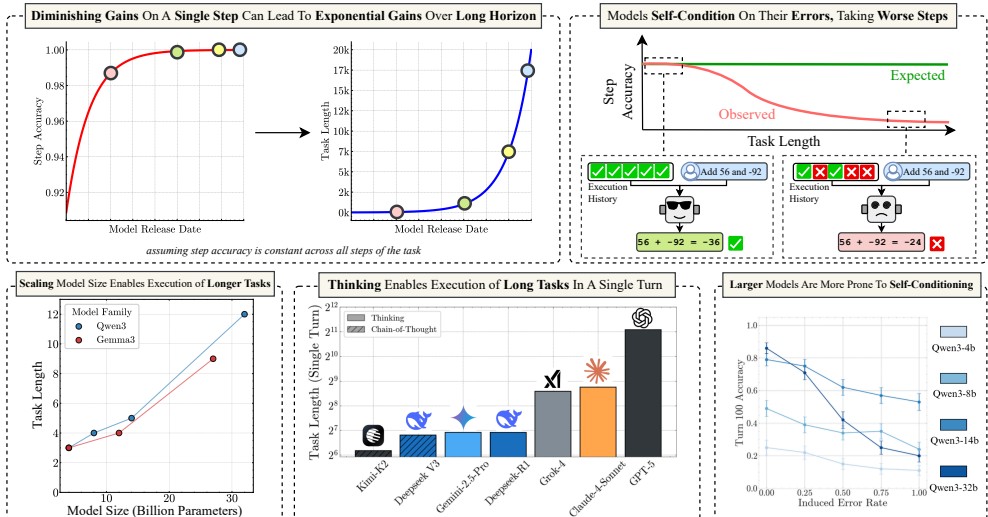

**Figure 1: A summary of our contributions.** Our work measures long-horizon execution, finding large benefits from scaling model size and sequential test-time compute. We identify a failure mode where models self-condition on their own errors, degrading future performance.

or tools to use and in which order, while *execution* involves carrying out the plan. In Shojaee et al. (2025) setup, the LLMs know the correct plan, as they initially follow it correctly for many steps. We posit that the eventual failures are in execution—as the task gets longer, the model is more likely to make a mistake in executing the plan. Although much attention has been paid to LLM planning abilities (Kambhampati et al., 2024), execution remains an understudied challenge, despite being increasingly important as LLMs begin to be used for long reasoning and agentic tasks.

In this work, we measure *long-horizon execution* capabilities of LLMs in a controlled setting. We isolate the *execution* capability of LLMs by explicitly providing them the *knowledge* and *plan* needed. By controlling the number of turns, and the number of steps per turn, which together contribute to task length, we reveal insights about long-horizon execution in LLMs:

**Does Scaling have Diminishing Returns?** We observe that diminishing improvements in single-step accuracy can compound, leading to exponential growth in the length of task a model can complete. Traditionally, scaling model size is assumed to increase capacity to store parametric knowledge or search for plans. Yet, even when the required knowledge and plan are explicitly provided, we find that scaling model size leads to large improvements in the number of turns a model can execute successfully.

**The Self-Conditioning Effect.** One might assume that failures on long tasks are simply due to the compounding of a small, constant per-step error rate and context length issues. However, we find that the per-step error rate itself rises as the task progresses. This is in contrast to humans, who typically improve at executing a task with practice. We hypothesize that, as a significant fraction of model training is to predict the most likely next token given its context, conditioning models on their own error-prone history increases the likelihood of future errors. We test this by controlling the error rate in the history provided to the model. As the error rate in the history is increased, we observe a sharp degradation in subsequent step accuracy, validating that models *self-condition*. We show how self-conditioning leads to degradation in model performance in long-horizon tasks beyond previously identified long-context issues, and unlike the latter, is not mitigated by scaling model size.

**The Impact of Thinking.** We find that recent thinking models are not affected by prior mistakes, fixing the self-conditioning effect. Further, sequential test time compute also significantly improves the length of task a model can complete in a single turn. Where, without chain-of-thought prompting, even frontier LLMs like DeepSeek-V3 fail at performing even four steps of execution, and its thinking version R1 can execute over 100 steps, highlighting the importance of reasoning before acting (Yao et al., 2023). We benchmark frontier thinking models, and find GPT-5 thinking (codenamed "Horizon") can execute over 2100 steps, far ahead of the next best competitor, Claude-4 Sonnet at 432.

The "jagged frontier" (Dell'Acqua et al., 2023) of LLM capabilities remains fascinating yet confusing. Unlike traditional machines, LLMs are more susceptible to failure when used for executing repetitive tasks. Thus, we argue that execution failures in long tasks should not be misinterpreted as the inability to reason. We show that long-horizon execution improves dramatically by scaling model size and sequential test time compute. If the length of tasks a model can complete indicates its economic value, continued investment in scaling compute might be worth the cost, even if short-task benchmarks give the illusion of slowing progress.

## 2 FORMULATION

First, we define key capabilities involved in an agentic or reasoning task. As a motivating example, consider an agent for the task of booking flights. Upon receiving a search result, the agent must reason to choose a flight that aligns with user preferences. One plan for this reasoning task could be:

For each flight, verify the flight timings, baggage allowance, and airline reviews. Then apply any available discounts or reward programs, and finally select a flight based on cost and travel time.

Each of these individual steps requires two operations: *retrieving* some information, and *composing* it with the existing information state, until the goal of choosing the final flight is reached. Both these operations require *knowledge*, potentially tacit, about how to perform them. *Execution* is carrying out this plan, a sequence of *retrieve-then-compose* steps, until a final booking is made. We formalize these terms for our work as follows:

> **Key Terms**
>
> **Planning.** Deciding what steps to take, and in what sequence.
>
> **Execution.** Carrying out the steps decided in the plan.
>
> **Knowledge.** Information about different types of steps, and how to compose them.
>
> **A reasoning task.** Requires planning the steps needed to solve it, and then executing them.
>
> **An agentic task.** Requires planning what actions to take, and then executing them.

In this work, we focus on *execution*, as we argue that it is a critical component of long-horizon capabilities. Execution has traditionally received less attention (Stechly et al., 2024) than capabilities such as reasoning, planning, and world knowledge, which have been the primary focus of LLM capability discussions. In fact, failures in execution have been misattributed to limitations in reasoning or planning capabilities (Shojaee et al., 2025; Khan et al., 2025). This perception may stem from the view that execution is straightforward or mundane. For example, once we as humans learn how to do a task, we are quite reliable at executing it, even improving with practice. However, as LLMs do not come with correctness guarantees, we posit that just execution over a long horizon can surprisingly be a challenge. We hypothesize that:

> *Even if planning and world knowledge are perfected,*
> *LLMs will still make mistakes in execution over a long-horizon.*

In an agentic or reasoning task, the model begins in an initial state (based on the first input) and has to perform a sequence of steps to reach the final goal. A long-horizon task requires a large number of steps, with the task length being the number of steps needed to complete it. We define the following metrics to evaluate performance:

> **Evaluation Metrics**
>
> **Step Accuracy.** Measures the fraction of samples where the state update from step $i - 1$ to step $i$ is correct, regardless of the correctness of the model's state at step $i - 1$.
>
> **Turn Accuracy.** A turn is a single interaction with the model, which may require executing multiple steps. Turn Accuracy measures the fraction of samples where the state update from turn $t - 1$ to turn $t$ is correct, regardless of the correctness of the model's state at turn $t - 1$.
>
> **Turn Complexity ($K$).** Defined as the number of steps the model has to execute per turn.

> **Task Accuracy.** Measures the fraction of samples in which the model can complete a task of $i$ steps without making any mistakes in the process.
>
> **Horizon Length ($H_s$).** Given a success rate threshold $0 \leq s \leq 1$, the horizon length is the first step $i$ where the model's mean task accuracy across samples drops below $s$. It can be interpreted as: the model can perform a task of length $H_s$ without making mistakes, with probability $s$. We use $s = 0.5$ unless otherwise specified, like Kwa et al. (2025).

## 2.1 DIMINISHING RETURNS IN STEP ACCURACY COMPOUND OVER A LONG HORIZON

We begin by analyzing the relationship between a model's single-step accuracy and its horizon length. Note that this analysis applies not just to execution, but rather to any general long-horizon task. To obtain a mathematical relation, we make two simplifying assumptions similar to LeCun (2023). First, we assume a model's step accuracy remains independent and constant over the task. Second, we assume a model does not self-correct, meaning any single error leads to task failure. We assume this only for the analysis here, which is illustrative and provides useful intuition. Our subsequent empirical analysis goes beyond this, investigating how LLMs, in fact, do not exhibit constant step accuracy for long-horizon execution, and may also correct their mistakes. Intuitively, if the model succeeds at one step with probability $p$, it's success probability after $t$ steps is $p^t$. This is often negatively viewed as errors compounding over a long task. But conversely, it also implies that a small improvement in step accuracy can lead to a large increase in horizon length after a certain (high) threshold of single-step accuracy is achieved.

**Proposition 1.** *Assuming an independent and constant step accuracy $p$ and no self-correction, the task-length $H$ at which a model achieves a success rate $s$ is given by:*

$$H_s(p) = \left\lceil \frac{\ln(s)}{\ln(p)} \right\rceil \approx \frac{\ln(s)}{\ln(p)}$$

*(The derivation is provided in Appendix J.)*

This shows that the horizon length grows hyperbolically with the step accuracy. We illustrate this growth in Figure 2 across different values for the success rate $s$. Notice the sharp growth in horizon length beyond 80% single-step accuracy, performance that frontier models now achieve on many question-answering benchmarks (Vendrow et al., 2025), which can be considered short tasks.

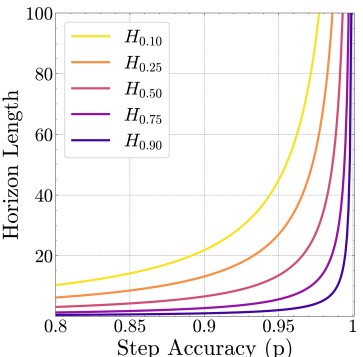

**Figure 2: Growth of Horizon Length.** The length of task a model can perform grows hyperbolically in the high accuracy regime.

We note that human labor is often compensated for its time. If the economic value of an agent also arises from the length of tasks it can complete, single-turn or short task benchmarks may be misleading for evaluating the benefits of further investment in LLM compute. While these benchmarks reveal genuine diminishing returns at the step level, they understate the compounding benefits that emerge over long-horizons. Beyond a threshold, small improvements in step accuracy can translate into success at rapidly increasing task lengths, which may provide a more faithful indicator of economic value.

For example, in METR's horizon length plot on software engineering tasks (Kwa et al., 2025), it was empirically observed that the horizon length at $s = 0.5$ of frontier models is growing exponentially, doubling every 7 months. Using our result above, in Figure 1 we show that such exponential growth in horizon length occurs even in a regime of diminishing returns on step accuracy. If we set $s = 0.5$, we obtain $H_{0.5} = -\frac{\ln(2)}{\ln(p)}$. As such, the step-accuracy $p$ required to sustain exponential growth in $H_{0.5}$ over time ($t$) is $2^{\frac{-1}{2^t}}$, which is indeed a diminishing function.

## 2.2 ISOLATING EXECUTION BY DECOUPLING PLANNING AND KNOWLEDGE

We now describe how we measure long-horizon execution empirically. We isolate execution failures by explicitly providing the requisite knowledge and plan. We study the chaining of the *retrieve-then-compose* step motivated in the flight-selection agent example earlier. Each step involves *retrieving*

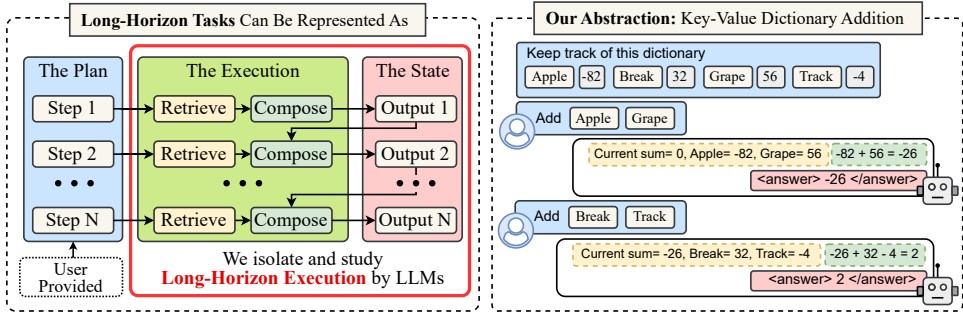

**Figure 3: Overview of our framework.** (Left) Our framework models long-horizon tasks as a sequence of *retrieve-then-compose* steps. (Right) We design a simple task that decouples planning from execution: in each turn, we provide the model the plan as key(s), asking it to *retrieve* their value(s), and *compose* them to maintain a running sum. We control the number of turns and turn complexity (keys per query).

relevant information or a tool specified by the plan and then *composing* its output to update the current state. The plan is deciding what to retrieve and how to compose it, whereas execution is actually performing those operations. This fits a natural abstraction—a key-value dictionary. The *key* serves as one step of a plan specifying what knowledge to retrieve, or tool to call, while the *value* represents the knowledge or tool output, which then has to be composed with the current state. In our study, we provide the plan as the keys in each query, eliminating the need for *planning* abilities from the LLM. We also provide the key-value dictionary in context, removing any dependency on the model's parametric *knowledge*. With this design, we directly control two important axes that multiply to obtain the task length (number of retrieve-then-compose steps): the number of turns, and the turn complexity ($K$). The turn complexity can be varied by changing the number of keys queried per turn.

## 3 EXPERIMENTS

We design a simple task where even language models with 4 billion parameters can achieve high accuracy, to isolate the capability of *long-horizon execution*.

**Setup.** As illustrated in Figure 3, we provide the model with the needed *knowledge*, a fixed, in-context dictionary $\mathcal{D} : \mathcal{V} \to \mathbb{Z}$, where $\mathcal{V}$ is a vocabulary of common five-letter English words and values are integers sampled uniformly from $[-99, 99]$. The initial state is $S_0 = 0$. In turn $t \in \{1, \dots, T\}$, the model receives an explicit *plan* $P_t = \{k_{t,1}, \dots, k_{t,K}\}$, which is a set of $K$ keys sampled from $\mathcal{V}$. For each turn $t$, the model must execute this plan, which requires updating the state, $S_t$, to maintain a running sum of values for all past queried keys. This requires the *retrieve-then-compose* steps:

1. **Retrieval:** Look up the integer value $\mathcal{D}[k]$ for each key $k \in P_t$

2. **Composition:** Sum these values and add them to the previous state, $S_t = S_{t-1} + \sum_{i=1}^{K} \mathcal{D}[k_{t,i}]$

We choose short English words and two-digit integers to minimize errors arising from tokenization. We provide few-shot examples to clarify the task. More details, including the exact prompt, are provided in Appendix G. We also disentangle performance on the individual retrieval and composition operations, finding that models have much higher accuracies on each of them alone (Appendix F).

### 3.1 EFFECT OF INCREASING THE NUMBER OF TURNS

We first test our hypothesis that long-horizon execution can be challenging even when a model has the required knowledge and planning ability, and then study the benefits of scaling model size.

**Setup.** We evaluate the Qwen3 (Yang et al., 2025a) and Gemma3 (Gemma-Team et al., 2025) model families, as they offer a range of sizes: [4, 8, 14, 32]B and [4, 12, 27]B parameters, respectively. For this experiment, we set the turn complexity to its simplest form ($K = 1$), providing a single key per turn, and vary the number of turns. Models are instructed to output the final answer directly, without intermediate thinking tokens, with the format enforced via few-shot examples. We verify that

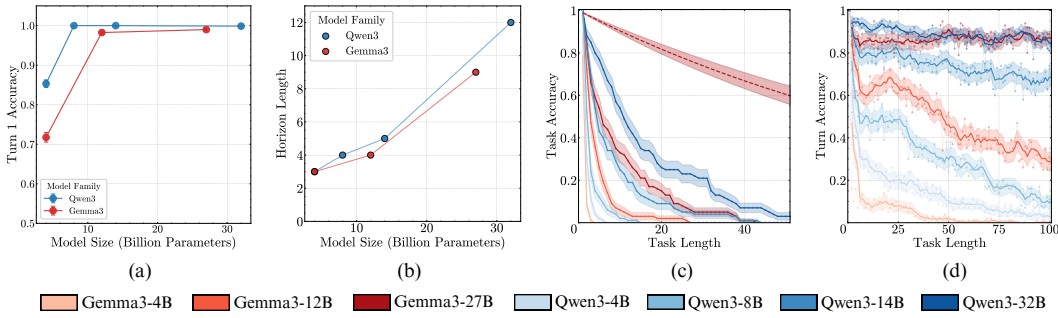

**Figure 4: Scaling model size has non-diminishing improvements in the number of turns it can execute.** The first-step accuracy for our task is near-perfect for all except the smallest models (a). Yet, as the model size is scaled, the horizon length increases significantly (b). We also see the effect of scaling in widening the gap between small and large models in task accuracy (c) and turn accuracy (d) as the number of turns increases. The shaded region is the mean $\pm$ one standard deviation over 100 samples; the solid line is the moving average over 5 turns; the dotted line is a hypothetical baseline model with constant step-accuracy of 0.99.

format-following errors are not the primary failure mode (Appendix H). We also show that the results below hold with chain-of-thought (CoT) prompting, and thinking models (Appendix Figure 13), and the trends are not affected by the temperature used (Appendix Figure 14).

**Result 1: Execution alone is challenging.** As seen in Figure 4 (a), all models except Gemma3-4B and Qwen3-4B achieve near-perfect accuracy on the first step, confirming they have the knowledge required to perfectly do a single step of our task. Yet, task accuracy falls rapidly over subsequent turns (Figure 4 (c)). Even the best-performing model (Qwen3-32B) sees its accuracy fall below 50% within 15 turns. This confirms our hypothesis that long-horizon execution can be challenging for LLMs, even if they have the needed knowledge and plan.

**Result 2: Non-diminishing benefits of scaling model size.** As shown in Figure 4 (c), larger models sustain higher task accuracy for significantly more turns, resulting in a clear scaling trend for horizon length (Figure 4 (b)). We abstain from deriving a "scaling law" since we can only obtain at most four model sizes from the same family, but the improvements do not seem diminishing. This observation is non-trivial. While the benefits of increasing model size are often attributed to improved capacity for knowledge, our task is not knowledge-constrained, as models achieve near-perfect first step accuracy (Figure 4 (a)), nor is the task more complex so that a larger model would be required. Yet, larger models are clearly more reliable at executing the task for longer. A possible explanation is the redundancy of internal circuits in larger models, which ensembles to reduce error (Lindsey et al., 2025). However, we find that simulating this redundancy with output-level aggregation of parallel compute does not replicate the gains observed from scaling model size (Appendix D).

### 3.2 WHY DOES TURN ACCURACY DEGRADE? THE SELF-CONDITIONING EFFECT

One might expect a model's turn accuracy to remain constant. Yet, Figure 4 (d) shows the accuracy of individual turns degrading as the number of turns increases. We investigate two competing hypotheses:

**1. Degradation as the context length increases.** The model's performance degrades simply due to increasing context length (Zhou et al., 2025a), irrespective of its content.

**2. Self-conditioning.** The model conditions on its own past mistakes. It becomes more likely to make a mistake after observing its own past errors in previous turns.

**Setup.** To disentangle these factors, we conduct a counterfactual experiment by manipulating the model's chat history. We control the error rate by injecting artificial output histories with a chosen error rate in the same format. If we fully *heal* the history, with a 0% error rate, degradation in the model's turn accuracy between turn 1 and a later turn can be attributed to long-context issues. If a model's accuracy for a fixed later turn consistently worsens with increasing error rate in prior turns, this would support our self-conditioning hypothesis.

**Result 3: Self-conditioning causes degradation in turn accuracy beyond long-context.** Our results in Figure 5 (a) show evidence for degradation due to both long-context and self-conditioning.

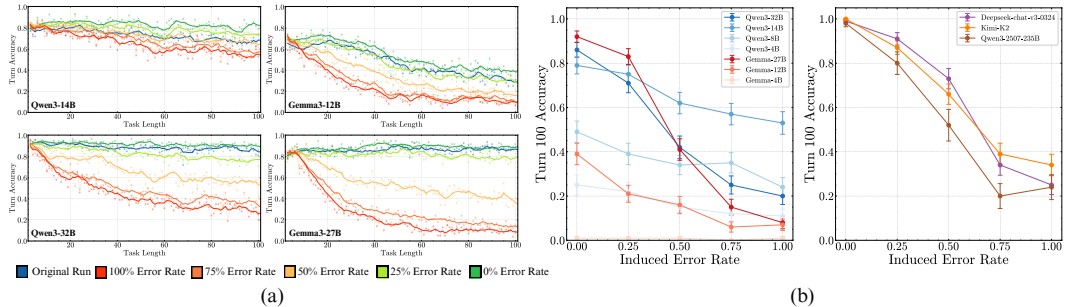

**Figure 5: Models self-condition on their previous mistakes, leading to more mistakes in subsequent turns.** By manipulating the chat history, we counterfactually vary the fraction of errors in previous turns. We find this increases the likelihood of errors in future turns (left). This shows a source of degradation in turn-wise model accuracy beyond long-context, as in the turn 100 slice (right) model accuracies are much higher when we provide a fully correct history. Scaling model size increases self-conditioning, even for frontier non-thinking models.

When conditioned on an error-free history (Induced Error Rate = 0.00), model turn accuracy at turn 100 is below its initial value, consistent with prior observations of long-context degradation (Zhou et al., 2025a). More interestingly, as we increase the rate of injected errors into the context, accuracy at turn 100 consistently degrades further. This demonstrates the self-conditioning effect—as models make mistakes, they become more likely to make more mistakes, leading to a continuous degradation in per-turn accuracy throughout the output trajectory as shown in Figure 5 (b).

**Result 4: Unlike long-context, scaling model size does not mitigate self-conditioning.** At the error rate of 0%, notice that the accuracy at turn 100 consistently improves as you scale model size. As shown in Figure 5 (b), scaling to frontier (200B+ parameter) models like Kimi-K2 (Kimi-Team et al., 2025), DeepSeek-V3 (DeepSeek-AI et al., 2025), and Qwen3-235B-Instruct-2507 (Yang et al., 2025a) largely solves long-context degradation for up to 100 turns, achieving near-perfect accuracy on a healed history. However, even these large models remain susceptible to self-conditioning, as their performance consistently degrades as the induced error rate in their history increases. A possible explanation for this is the relation between self-conditioning and in-context learning. In in-context learning, the model desirably conditions on user and environment inputs, for example, the correct few-shot demonstrations of the task we also provide in the initial prompt. In contrast, in *self*-conditioning, the model conditions on its *own* past behaviour, which could be wrong, leading to more mistakes. At a fundamental level, both arise from the model trying to output the most likely completion to its context, which larger models are known to be better at (Arora et al., 2025). Self-conditioning is also consistent with results showing how hallucinations snowball (Zhang et al., 2023), and larger models shift more in personality during multi-turn conversations (Choi et al., 2024; Becker et al., 2025), where in our case, the drift is toward a personality that makes errors.

In Appendix I, we try the above setup of output manipulations with CoT prompting, finding that accuracy still deteriorates as the induced error rate increases. A potential confounder is that the manipulated outputs deviate from the CoT. We try to mitigate this issue with programmatically generated CoT traces, but still observe self-conditioning. We also try removing CoT traces from previous turns from history, which causes the model to stop using CoT completely. So, we now present results for the Qwen3 thinking models, which are trained with reinforcement learning (RL) to think even when previous turn traces are not presented. As before, we observe their turn 100 accuracy while controlling the error rate in prior turns.

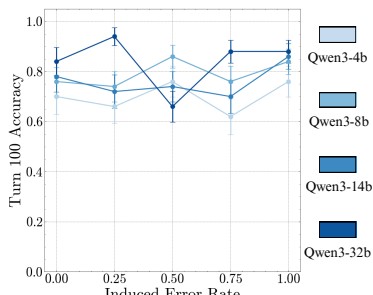

**Figure 6: Thinking fixes self-conditioning.** Qwen3 models with thinking enabled no longer self-condition, even when the entire prior history has wrong answers, in contrast to non-thinking results.

**Result 5: Thinking fixes self-conditioning.** In Figure 6, we observe that the Qwen3 thinking models do not self-condition—the accuracy of the models at turn 100 remains stable, regardless of the error

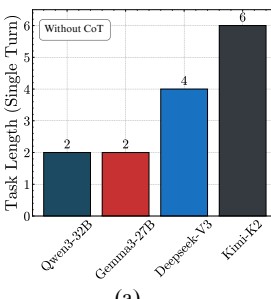 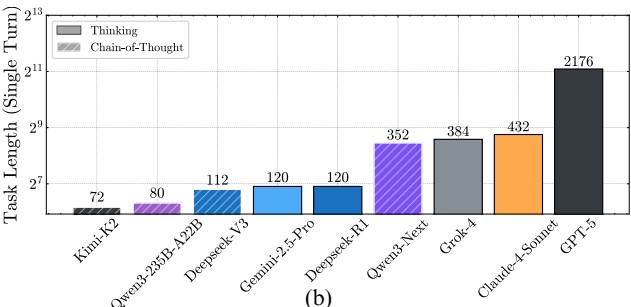

(a)                                    (b)

**Figure 7: Benchmarking the length of task models can execute in a single turn.** Without CoT or thinking, even the biggest models fail to execute more than a few steps (a). Sequential test time compute (thinking tokens) significantly improves this, especially when trained with RL (b), where GPT-5 is far ahead of the rest.

rate in its context. This could arise from two reasons. First, RL training can reduce the most likely next token prediction behaviour of language models, making them oriented towards task success rather than continuing the context. Second, the removal of thinking traces from prior turns could reduce the influence of prior turns on the model's output, as it thinks about the new turn independently. By inspecting the models' thinking traces, we observe that they do not refer back to their answers in prior turns, which could be a potential reason why they do not self-condition. Inspired by this, for non-thinking models, we experiment with context engineering by explicitly removing prior history and find that it indeed mitigates self-conditioning (Appendix C.2). We also find that just prompting models to self-verify their answers does not solve self-conditioning completely (Appendix C.1).

### 3.3 WHAT IS THE LENGTH OF TASKS MODELS CAN COMPLETE IN A SINGLE TURN?

In the previous sections, we measured how many turns models can successfully execute a single retrieve-then-compose step. However, most real-world tasks require more complex processing every turn. In fact, in Appendix E we show that the horizon length of different models can vary significantly at different turn complexities. The total task length a model can handle is a function of both the number of turns and the number of steps to execute per turn. We now measure the latter dimension: the maximum number of steps a model can execute per turn.

**Setup.** To quantify the length of task models can complete in one go, without user input, we run a binary search (Lehmer, 1960) to find the highest turn complexity ($K$, the number of keys) the model can provide the correct sum for with accuracy $\geq 80\%$. We evaluate a suite of frontier models like GPT-5 (OpenAI, 2025), Claude-4 Sonnet (Anthropic, 2025), Grok 4 (xAI, 2025), Gemini 2.5 Pro (Gemini Team, 2025), Kimi K2 (Kimi-Team et al., 2025), and DeepSeek-R1 (Guo et al., 2025). An advantage of our benchmark is that it is contamination-free, as new examples can be generated programmatically.

**Result 6: Without CoT, non-thinking models struggle to chain more than a few steps per turn.** In Figure 7 (left), we first find that when prompted to answer directly, without chain-of-thought, the larger Qwen3 32B, Gemma3 27B, as well as frontier non-thinking models like DeepSeek-V3 (670B), and Kimi K2 (1026B), fail to execute even a turn complexity of more than six. This is consistent with prior work showing the necessity of thinking tokens for transformers to perform sequential tasks (Weiss et al., 2021; Merrill & Sabharwal, 2023). We see that the number of steps the model can execute in a single turn improves significantly with chain-of-thought. This reinforces the importance of reasoning before acting (ReAct (Yao et al., 2023)) for agents, even if this costs more and fills up the context window. We show preliminary evidence that parallel test time compute is not as helpful, with majority voting leading to only marginal improvements in execution length (Appendix D).

**Result 7: Benchmarking frontier models.** In Figure 7 (right), we benchmark frontier models on the length of task they can execute in a single turn. We find a surprisingly large gap between GPT-5 (codenamed *Horizon*) with 2176 steps and others like Claude-4 Sonnet (432 steps), Grok 4 (384 steps), and Gemini 2.5 Pro (120 steps). Qwen3-Next, which combines Gated DeltaNet (Yang et al., 2025b) with standard attention, designed explicitly for long-context tasks, outperforms even larger models that use only standard attention, hinting at the impact of model architecture for long-horizon

tasks. Overall, even our simple task can separate frontier models in their long-horizon execution capability, and presents a clear opportunity to improve current open-weight models.

## 4 RELATED WORK

**Increasing Task Length.** Multiple works have recently shown how models worsen as *problem complexity* increases (Zhou et al., 2025b), often attributed to failures of reasoning (Cheng, 2025; Shojaee et al., 2025). Recently, multiple real-world long-horizon agentic benchmarks have been proposed (Backlund & Petersson, 2025; Xie et al., 2024; Shen et al., 2025), where prior work has studied planning failures (Chen et al., 2024b). By designing a task where no reasoning is required, given that we provide the model the requisite plan and knowledge, we show that execution alone can be a challenge, degrading model accuracy on longer tasks. Our observations on scaling could hold for the related problem of length-generalization—training models to succeed on tasks longer than those seen during training (Fan et al., 2024; Cai et al., 2025).

**Long Context.** Much of prior work has focused on improving the maximum context length that can be provided in the input to a language model (Su et al., 2021), and evaluating whether (Tay et al., 2020) and how (Olsson et al., 2022; Li et al., 2023) models maintain performance as the context gets longer (Tay et al., 2020). Closest is the recent RULER (Hsieh et al., 2024) and GSM-Infinite (Zhou et al., 2025b), which also use synthetic data to systematically evaluate long-context abilities. While long-context will help models execute for longer, it is a different capability compared to long-horizon execution (Zhou et al., 2023; Chen et al., 2024a), as it focuses on performance as a function of input, not output length. We identified one such difference, the self-conditioning effect—where past errors in model output increase the chance of future mistakes, and disentangle this effect from long-context degradation in Section 3.2.

**Self-correction.** To address an issue similar to self-conditioning at the token level called "exposure bias", Ranzato et al. (2015) proposed using reinforcement learning to train language models. This trains the model to react to mistakes in its prior outputs by correcting them, instead of making more of them, an intuition that also guides more recent work on RL for self-correction (Kumar et al., 2024; Ma et al., 2025). As a training-free fix for our setting, we also try model-based self-verification (see Kamoi et al. (2024) for a survey) at each turn in Appendix C.1, which can be considered a form of self-refinement (Madaan et al., 2023) without having access to environment feedback (Gou et al., 2023).

**Scaling LLMs and RL.** Scaling laws for language models show diminishing returns on the loss for the single step of predicting the next token (Kaplan et al., 2020; Hoffmann et al., 2022). When models competed in simple knowledge-based question-answering tasks such as MMLU (Hendrycks et al., 2020), such single-step measurements could inform us about the rate of progress. This has changed in the last year. Where earlier we could only post-train on human demonstrations (Mishra et al., 2021), language models can now be trained with just rewards (Shao et al., 2024), enabling sophisticated reasoning (Guo et al., 2025; Jain et al., 2024) and agents (Kimi Team et al., 2025). This opens up the opportunity to solve much longer tasks where earlier human supervision would be too expensive to scale. Our work shows how diminishing returns on single-step performance can compound to provide large benefits in the length of tasks a model can solve. This motivates the need to study empirical scaling laws for horizon length in agents (Hilton et al., 2023).

**Tool Use.** In symbolic AI, once tasks are formalized, for example, into STRIPS plans (Fikes & Nilsson, 1971), they can be executed without issues. Prior work (Chen et al., 2024b; Valmeekam et al., 2024) has shown LLMs struggle to match symbolic algorithms for automated planning. In contrast, we show LLMs can fail on straightforward execution (Zhu et al., 2025b; Sun et al., 2025; Stechly et al., 2024) over a long horizon even when the plan is provided. Teaching LLMs to use tools offers one way to shift the burden of execution from probabilistic models to reliable programs (Schick et al., 2023). However, reasoning is often fuzzy and not always easy to implement as a tool, requiring the model to execute some steps by itself. Even calling the right tools requires reliable execution from the model (Patil et al., 2025).

## 5 CONCLUSION

In this work, we show how short-task benchmarks may give the illusion of slowing progress for modern language models. We show that scaling model size increases the number of turns a model

can execute, while sequential test-time compute increases the length of tasks a model can perform on a single turn. Together, these contribute to dramatically increasing horizon lengths for LLMs.

**Limitations.** As with any "synthetic" task (Allen-Zhu, 2024; Poli et al., 2024; Chollet et al., 2024) used for a controlled study of LLM capabilities, there are a few limitations of our setup. Improvement on our task is necessary, but not sufficient for long-horizon execution on real-world tasks. It does not reflect the complexities and sources of error arising in real agentic tasks with a large number of possible actions. In such settings, the number of actions and the accuracy of each action can both vary based on the plan, requiring more careful consideration. Another issue in more complex tasks could be the possibility of multiple distinct correct plans, such that even if we provide a plan to the model, it could deviate from it while being correct. This is not possible in our controlled study, as using the wrong keys would likely lead to a wrong outcome. In Appendix A, we demonstrate how self-conditioning is a significant failure mode for LLM agents even in realistic benchmarks like ALFWorld (Shridhar et al., 2021), GAIA (Mialon et al., 2024), and WebShop (Yao et al., 2022). Our results are observations about current LLMs, and not inherent properties of transformers, so they might change with task-specific finetuning. Particularly, the single-turn task in Section 3.3 is in theory parallelizable, which we discuss further and propose pathways to future-proofing via more inherently sequential tasks in Appendix B. Finally, our current task accuracy metric does not account for self-correction. In tasks where mistakes are acceptable and easy to undo, self-correction is a promising direction to improve long-horizon execution.

**Outlook.** Scaling up the length of tasks a model can complete would be a major step towards realizing the true potential of general, open-ended agents (Raad et al., 2024). If they are trained in simulated environments created with generative models (Bruce et al., 2024), maintaining accuracy over a long-horizon becomes doubly important. By showing long-horizon execution can be studied on simple tasks, we hope to inspire more research on this capability, as it is an increasingly important capability in the era of experience (Silver & Sutton, 2025).

## REPRODUCIBILITY STATEMENT

To ensure the reproducibility of our findings, we have provided comprehensive details of our methodology, experiments, and theoretical results. The mathematical derivation and assumptions for Proposition 1 are detailed in Appendix J. All specifics of our experimental methodology are consolidated in Appendix G, which covers our synthetic task design and data generation (Appendix G.1), the exact prompts used for both standard and thinking models (Appendices G.2 and G.3), model specifications and hyperparameters like temperature and top-p (Appendix G.4), and the computational resources used for the evaluations (Appendix G.5). Furthermore, we have included the complete source code, data generation scripts, and the exact datapoints used as supplementary material.

## ACKNOWLEDGEMENTS

We thank Maksym Andriushchenko, Nikhil Chandak, Paras Chopra, Dulhan Jayalath, Abhinav Menon, Sumeet Motwani, Mathias Niepert, Lluís Pastor Pérez, Ameya Prabhu, Tim Schneider, Shashwat Singh, and Timon Willi for helpful feedback. AA was funded by the CHIPS Joint Undertaking (JU) under grant agreement No. 101140087 (SMARTY), and by the German Federal Ministry of Education and Research (BMBF) under the sub-project with the funding number 16MEE0444. AA thanks the International Max Planck Research School for Intelligent Systems (IMPRS-IS) and the ELLIS PhD programs for support. The authors gratefully acknowledge compute time on the Artificial Intelligence Software Academy (AISA) cluster funded by the Ministry of Science, Research and Arts of Baden-Württemberg.

## AUTHOR CONTRIBUTIONS

SG conceived the project. AS led the execution of the experiments with the help of AA, while SG led their planning with the help of AA, AS, and JG. SG and AA wrote the paper, while AS worked on the figures. JG and SS advised the project, providing valuable feedback throughout.

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

# Appendix

CONTENTS

## A  SELF-CONDITIONING IN REALISTIC AGENTIC TASKS

Zhu et al. (2025a) recently released[1] annotated failed trajectories from more realistic, popular agent benchmarks like GAIA (Mialon et al., 2024), ALFWorld (Shridhar et al., 2021), and WebShop (Yao et al., 2022). We manually analyzed each annotated sample and identified potential instances of self-conditioning. We attach an example of an annotated entry below.

```
"trajectory_id": "GPT-4o_001_chat_b000_t00_e00-d3248f80",
"LLM": "GPT-4o",
"task_type": "webshop",
"critical_failure_step": 29,
"critical_failure_module": "plan",
"step_annotations": [
  {
    "step": 29,
    "plan": {
      "failure_type": "inefficient_plan",
      "reasoning": "This plan loops back to the very beginning, moving
          in the wrong direction, and results in an inefficient search
          query that is almost identical to the initial ones."
    }
  }
]
```

We search for entries where the failure reason mentions the agent repeating the previous mistakes repeatedly and eventually degenerating into failure trajectories. Some of the error categories that fit our search were,

- `inefficient_plan`: Plan is overly long or illogical because of prior errors.
- `progress_misjudge`: Incorrectly evaluates progress based on its own prior outputs.
- `causal_misattribution`: Correctly notes failure but blames the wrong cause due to its prior outputs, misguiding subsequent plans.

Through this process, we estimate that roughly 20% of GAIA, 48% of ALFWorld, 33% of WebShop failures are similar to self-conditioning. Here are some such trajectory annotations as examples:

**WebShop:** *"...This set the agent on an endless loop of query reformulation and paging through irrelevant results, making success impossible..."*

**ALFWorld:** *"...This set a precedent for all subsequent memory outputs to ignore key details about which objects had been visited what actions were performed at each and what the actual observations were..."*

**GAIA:** *"Make an low efficient plan that not success, repeat the similar action that not success"*

A caveat in the analysis we present above is that the correctness of steps is subjective to determine on these tasks for a quantitative study of self-conditioning. This is exactly where having a controlled study like ours, even if with a synthetic task, is extremely valuable. It allows us to isolate execution failures from planning and knowledge gaps, which is difficult in more complex agentic tasks.

## B  SEQUENTIALITY OF THE SINGLE-TURN TASK

A potential limitation of the single-turn setting in Section 3.3 is that retrieving and adding a list of numbers is, in theory, parallelizable inside a transformer (Merrill & Sabharwal, 2023). This is

---

[1]Link to AgentErrorBench trajectories

because adding a list of numbers lies in the Threshold Circuit 0 (TC0) (Hajnal et al., 1993) class of problems that can be solved with constant depth computation. While currently our results show that frontier language models fail to perform more than a few such additions without chain of thought, since this is in theory learnable, our single-turn study is not future-proof. Note that this limitation does not apply to our multi-turn experiments, as the keys to be retrieved and their value added is provided turn-by-turn, and thus the model has to perform these steps sequentially. To mitigate this, we now perform a study with a sequence of $2 \times 2$ (non-stochastic) matrix multiplications, which in the general case lies in TC1, generally believed to not be equivalent to TC0 (Mereghetti & Palano, 2000).

**Experimental Setup.** To keep this task similar to the key-value addition task introduced in Section 3, we create an analogous key-value matrix multiplication task, where each key corresponds to a randomly generated $2 \times 2$ matrix. Since multiplication causes values to blow up to very large numbers quickly, we ensure each element in the matrix is in the range of $[-9, 9]$. The LLM is then prompted to complete a chain multiplication of $K$ matrices, where $K$ corresponds to the per-turn complexity (Section 3.3). The full prompt is presented below. Analogous to the key-value addition task, we again have a series of *retrieve-then-compose* steps:

1. **Retrieval:** Look up the matrix $\mathcal{M}[k]$ for each key $k \in P_t$

2. **Composition:** Compute the product of these matrices and multiply by the previous state, $S_t = S_{t-1} \cdot (\prod_{i=1}^{K} \mathcal{M}[k_{t,i}])$

```
Starting Prompt
You are an AI assistant.  I will provide you with a dictionary of keys mapped to 2x2
matrices and give you keys in groups of 2.  Maintain the running 2x2 product starting
from the identity matrix.
For each group, respond with the current product in row-major order inside
<answer></answer> tags using four comma-separated integers.
IMPORTANT: Do not output any other text inside <answer> tags.  Do NOT output anything
else at all.

Illustrative examples using a reference dictionary:

alpha:  [[1, 0], [0, 1]]
beta:  [[2, 1], [0, 1]]
gamma:  [[1, 0], [1, 1]]
delta:  [[0, 1], [1, 0]]

Example 1:
Input Keys:  ['alpha', 'beta', 'gamma']
Illustrating with product after each key:
Result:  <answer>1,0,0,1 | 2,1,0,1 | 3,1,1,1</answer>

Example 2:
Input Keys:  ['delta', 'beta', 'gamma', 'alpha']
Illustrating with product after each block of 2 keys:
Result:  <answer>0,1,2,1 | 1,1,3,1</answer>

Example 3:
Input Keys:  ['beta', 'gamma', 'beta']
Illustrating with product after each block of 3 keys:
Result:  <answer>6,4,2,2</answer>

Now, here is the actual task:
Dictionary to maintain:

drive:  [[-3, 9], [1, -3]]
alone:  [[6, 3], [5, -5]]
adult:  [[-1, -5], [-2, 8]]...
Ready to start!
IMPORTANT: Only output the four comma-separated integers representing the matrix product
inside <answer> tags.  Nothing else should be present in the output, ONLY the answer.
```

**Results.** Our findings for the iterated matrix multiplication task, presented in Figure 8, are similar to those presented in Figure 7. For each model, we evaluate 50 independent rollouts. Without sequential test-time compute, models are unable to multiply even 2 matrices. This is expected, as each step of this task requires a constant number of sub-tasks, including multiple retrievals, multiplications, and

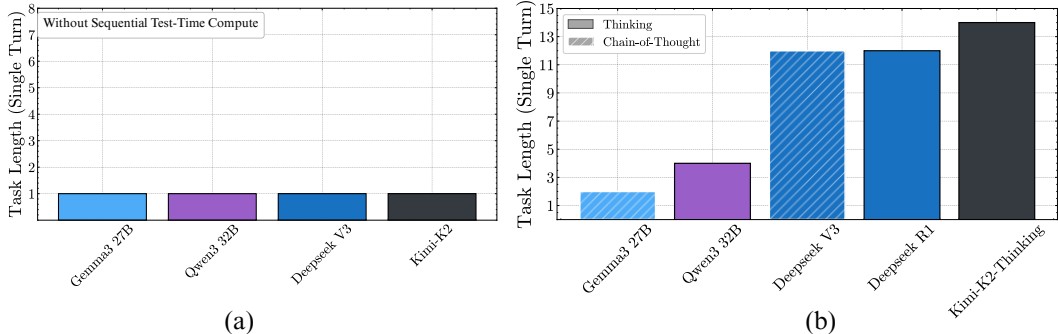

**Figure 8: Benchmarking iterative matrix multiplication within a single turn.** Without CoT or thinking, even the biggest models fail to multiply just two matrices (a). Sequential test time compute (thinking tokens) significantly improves this (b).

additions. With sequential test-time compute, we see a substantial increase in each model's ability to perform the task.

## C  INVESTIGATING POTENTIAL FIXES FOR SELF-CONDITIONING

### C.1  TURN-WISE SELF-VERIFICATION PROMPTING

We investigate whether the self-conditioning effect can be mitigated by explicitly prompting the model to perform active self-correction. At each turn, we instruct the model to first re-validate its previously reported state and, if required, recalculate the full historical sum before processing the current turn's keys.

The results, shown in Figure 9, are mixed. For the Gemma3 family with CoT, this setup provides an initial boost in accuracy, successfully breaking the self-conditioning loop in early turns. However, the self-verification process significantly increases the number of tokens generated per turn, causing the model to exhaust its context window much sooner, which leads to a sharper performance collapse in later stages. In contrast, the Qwen3 thinking models show negligible improvement. Manually inspecting their reasoning traces, we find that these models, likely due to their fine-tuning, overthink and frequently fail at the verification step itself, sometimes making arithmetic errors even during their re-calculation process.

These findings suggest that prompting-based self-correction may not be enough. It is computationally expensive, incurring a context-length penalty, and is itself a complex, error-prone execution task that models may not be able to perform reliably.

### C.2  CONTEXT ENGINEERING

Another natural mitigation strategy is to limit the model's exposure to its own past errors in its history. We operationalize this using a simple sliding context window, which is particularly well-suited for Markovian tasks like ours. This approach maintains only the $N$ most recent turns in the model's context. The rationale is that a smaller context window reduces the probability of the model observing a lot of its own past failures, thereby breaking the negative feedback loop of self-conditioning.

As shown in Figure 10 (a), performance improves significantly as the context window size is reduced, allowing models to sustain execution for longer horizons. While a fixed sliding window is only applicable to tasks without long-range dependencies, this result validates a more general principle: active context management designed to minimize the accumulation of errors in the context is a promising direction for improving long-horizon reliability in LLM agents.

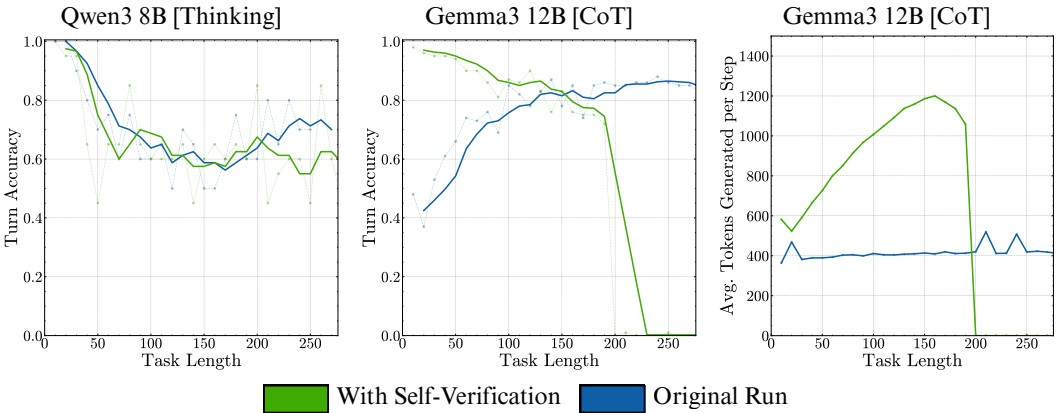

**Figure 9: Self-verification does not fix self-conditioning**. Prompting to self-verify does not suffice to fix the self-conditioning effect completely. It leads to overthinking in thinking models and increases the amount of tokens required per turn, leading to faster context consumption in CoT models.

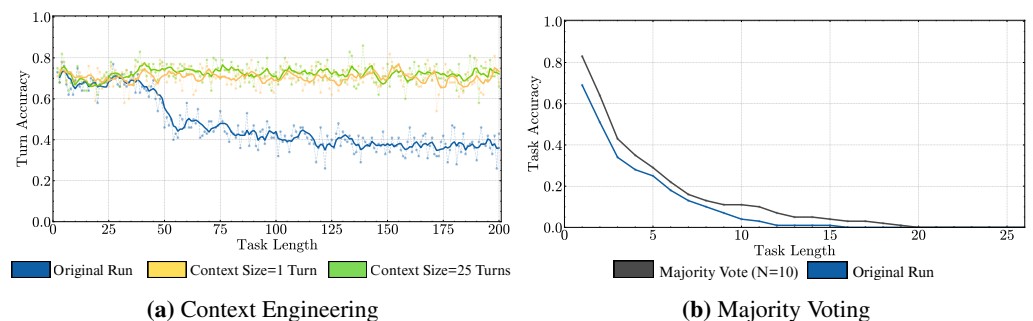

**(a)** Context Engineering  **(b)** Majority Voting

**Figure 10: Context engineering and majority voting on Gemma3 12B.** Controlling context size reduces the self-conditioning effect, but relies on the Markovian nature of our task. Majority voting at K=1 provides only minimal improvements over the baseline.

## D  CAN PARALLEL TEST-TIME COMPUTE SCALING MATCH THINKING?

We also experiment to validate if parallel scaling in test-time compute can achieve the same improvements as thinking. We verify this by testing if parallel majority voting can replicate the gains from either model scale or sequential computation (*thinking*). To create a fair comparison, we sample multiple outputs from a non-thinking Gemma3 model at each turn, with the number of samples set to match the average token count of its CoT counterpart. The final answer is determined by a majority vote over these parallel generations. From the results in Figure 11 and 10 (b), we see that while majority voting yields a marginal performance improvement over the base model, it is insufficient to match the reliability of a larger, non-thinking model, let alone the substantial gains from using CoT reasoning. This suggests that for long-horizon execution, sequential computation provides an advantage that parallel test time scaling cannot match. This contrasts with findings in other domains, such as math or common-sense reasoning, where parallel sampling with self-consistency has been shown to be highly competitive (Snell et al., 2024).

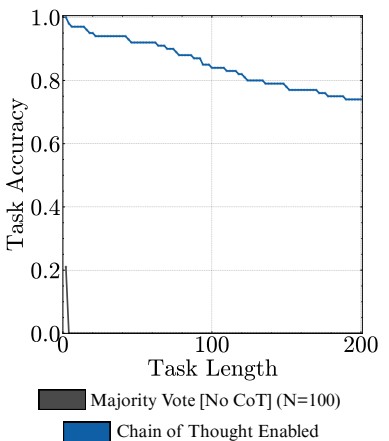

**Figure 11: Parallel test time scaling on Gemma3 12B at K=2.** Majority voting with the same amount of tokens as CoT traces does not match the performance of CoT.

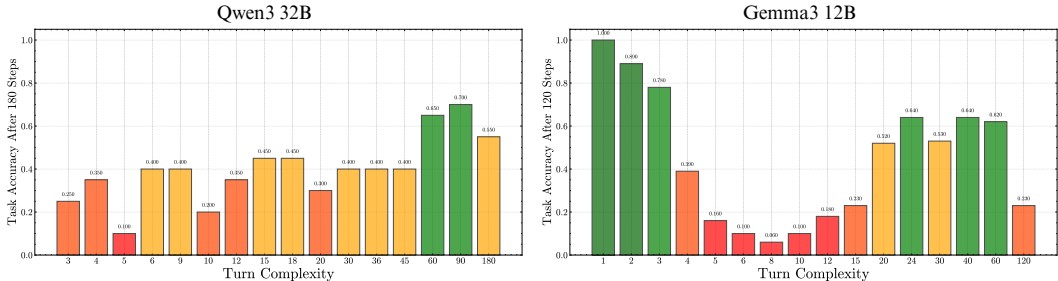

**(a)** For the same number of total steps, different turn complexities lead to different outcomes. We find no trend across families.

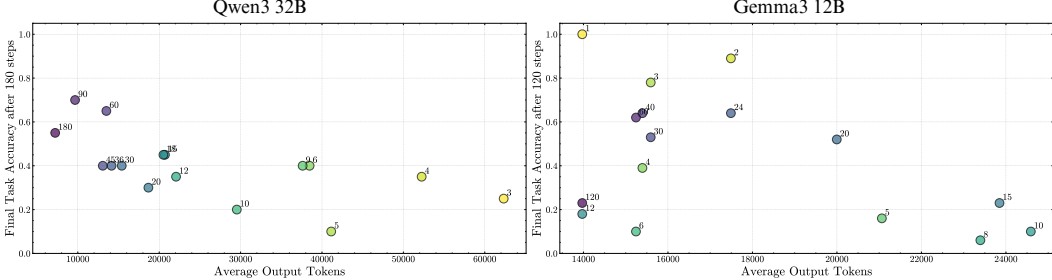

**(b)** Average output tokens used to complete the execution vs final accuracy. We see that for Qwen3 32B, more turns lead to more token usage, even at lower turn complexities, pointing to overthinking. Gemma3 12B, on the other hand, uses fewer tokens for very low turn complexity or very high turn complexity.

**Figure 12: Relation between the turn complexity and the number of turns.**

## E  NUMBER OF TURNS VS TURN COMPLEXITY

In our experiments, we show that we can increase the length of the task needed to be performed by either (1) increasing the number of turns or (2) increasing the turn complexity, i.e, providing more inputs in the same turn. To investigate the relationship between these two axes, we perform an experiment where a model has to perform a fixed number of operations while varying the turn complexity. With a higher turn complexity, the model requires fewer turns to reach the fixed number of operations. Results in Figure 12 (a) indicate there is no strict turn complexity that is consistently the best across model families. Rather, we found that different models behave quite differently for the same turn complexities. Qwen3 32B seems to show poorer performance at lower turn complexities, indicating that it is unable to perform well over a large number of turns, even if the turns are simple themselves. Gemma3 12B shows a different trend. It reaches accuracy peaks at either extreme of the turn complexity spectrum, failing badly at mid-level turn complexities. This indicates it suffers when the turn complexity and the number of turns are both sufficiently high.

Another axis of evaluating the number of turns vs turn complexity trade-off is the test-time compute used. From a cost perspective, increasing the number of turns increases the overall cost of inference. We can lower the number of turns by increasing the turn complexity, but that would result in an increase in the per-turn inference cost, as a result of the added complexity. For the same experiment above, we track the number of output tokens used for computation (including thinking tokens) per sample, and again find diverging results for each family in Figure 12 (b).

## F  DECONSTRUCTING ERRORS IN RETRIEVE-THEN-COMPOSE STEPS

To further isolate the source of execution errors, we decompose our task into its two constituent operations–retrieval and addition–and evaluate models on them individually:

- **Retrieval-Only Task.** A stateless task where, at each turn, the model is given a key and must simply return the corresponding integer value from the dictionary. No running sum is maintained. This isolates the retrieval component.

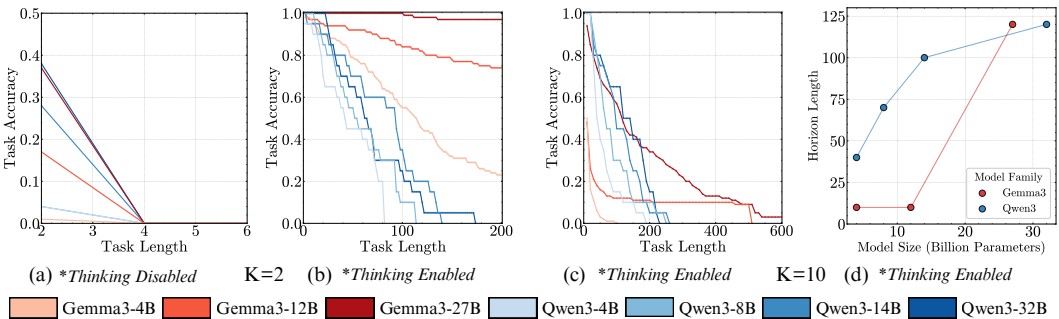

**Figure 13: Scaling trends hold even after enabling Sequential Test Time compute.** We compare model performance with thinking disabled (a) against thinking enabled for Qwen models and CoT for Gemma models (b, c) at varying turn complexities. (a) Without thinking, all models fail to execute even two steps ($K = 2$) in a single turn. (b) In contrast, enabling thinking prevents this performance collapse, with all models successfully handling $K = 2$. (c) When the turn complexity is further increased to $K = 10$, performance degrades, but a clear scaling trend emerges. (d) This trend is explicitly shown, illustrating that for complex turns, the horizon length increases consistently with model size, reinforcing the benefits of scaling model size even when thinking is enabled.

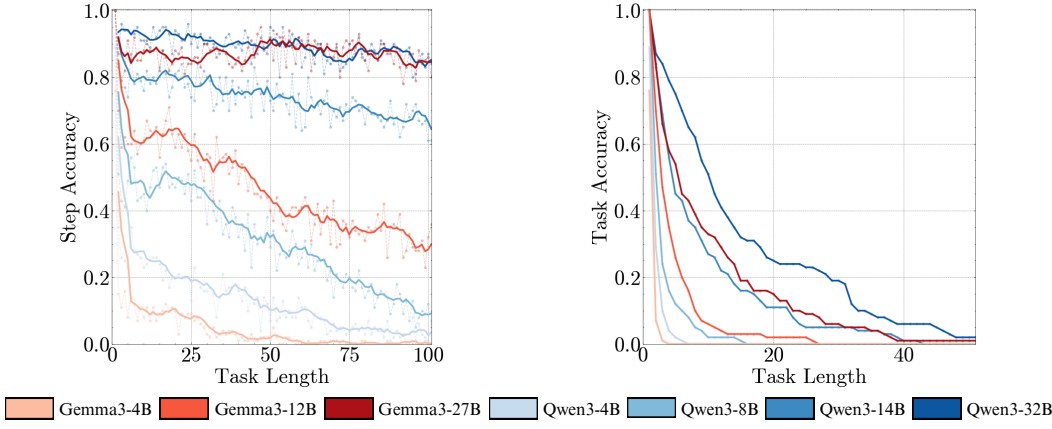

**Figure 14: Temperature does not impact the trends observed.** We reproduce the same trends in Figure 4, when running with temperature 0.

- **Addition-Only Task.** A stateless task where, at each turn, the model is given two random integers to add. No running sum is maintained. This isolates the arithmetic component.
- **Prefix-sum Task.** A stateful task where, at each turn, the model is given an integer directly and must add it to its previously reported running sum. This isolates the combination of arithmetic and state-tracking components.

From Figure 15, we can observe that models achieve near-perfect performance on the stateless retrieval and addition task, indicating that neither simple dictionary lookup nor addition is a significant source of error. In contrast, the prefix sum task, while significantly better than our task, still exhibits a slow degradation over time.

This leads to two key insights. First, the difficulty lies not in the atomic operations themselves, which models perform with high accuracy in isolation over long horizons. Second, this suggests that the primary source of degradation is the *state-management* component of the task. While stateless retrieval and addition are trivial, the requirement to reliably maintain and update a running sum introduces higher chances of error. This suggests that the models struggle with the requirement to concurrently manage information lookup and state updates.

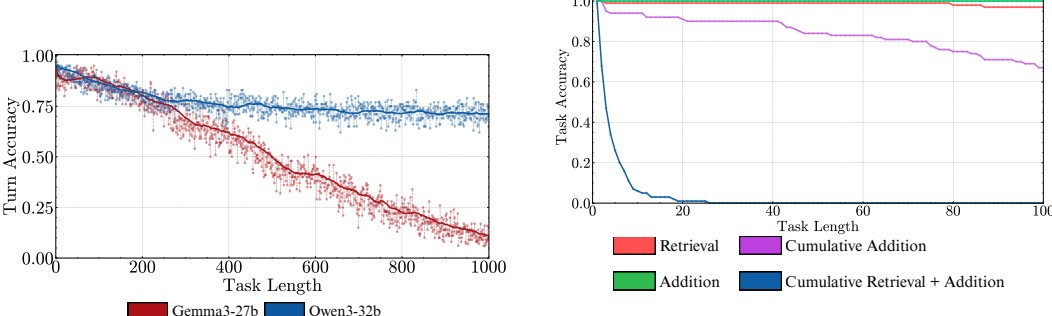

**Figure 15: Analysis of execution failures.** (a) Self-conditioning effect emerges as tasks get longer. Even for models that ace the task at a task length of 100, the Turn Accuracy drops constantly as we further increase the turns. (b) Models are good at the tasks individually, but not on their composition. State tracking introduces additional difficulty.

# G  EXPERIMENTAL SETUP

## G.1  TASK DETAILS

We create a dictionary where keys consist of common five-letter English words, and the values consist of integers uniformly sampled from $-99$ to $99$. The range of values is deliberately kept large to minimize the chance of an assistant being wrong in an earlier turn correcting its response by pure accident. For our experiment, we first create a fixed set of 100 keys. Then, we create multiple rollouts (samples) of $50,000$ steps. For each rollout, we uniformly sample a separate set of values to be assigned to each key, in order to increase experimental breadth. Next, at each step, we uniformly sample a key to be provided at that step with replacement. This gives us a list of keys to be processed in order, which is exactly the plan to be executed by the agent.

To account for the turn complexity ($K$), we group $K$ consecutive keys together and represent them as *one turn*. Thus, we can fully specify our intended evaluation by (1) specifying the number of samples (rollouts) needed, (2) the turn complexity, $K$, and (3) the number of turns required. This gives us a superset of data from which we sample rollouts to use in our experiments. For the Qwen3 and Gemma3 families, we sample 100 rollouts. For frontier models, due to cost limitations, we sample 20-50 rollouts. To ensure consistency in evaluation, we provide the same rollouts to each model.

## G.2  PROMPTING

Each LLM is provided a standardized prompt describing the task at the start of the conversation. This prompt specifies the dictionary containing the five-letter word keys and their corresponding values.

Further, the prompt specifies the number of keys that will be provided to the LLM at each subsequent turn. To ensure format following, the prompt also contains few-shot examples with different turn complexities. Finally, the LLM is asked to provide the running sum after each turn in `<answer>` tags. An example conversation is shown below.

## G.3 PROMPTING FOR THINKING MODELS

To enable models to use chain-of-thought prompting, we add the line "`Think step by step before answering.`" to the prompt and also add CoT traces to the few-shot examples. We find that models stop performing CoT reasoning after a few turns, as it starts conditioning on the answer format in its history. Thus, we end up *including the chain-of-thought trace* in the conversation history, to ensure the model does not forget the CoT instruction. This is a trade-off we had to make as it increases the input context of the LLM, however, it was essential to ensure instruction following. Thinking models provided their reasoning in `<think>` tags, which were removed from the conversation history. No other changes were needed to make the thinking models follow instructions.

## G.4 MODEL SPECIFICATIONS

For chain-of-thought prompting, we set the per-turn output token limit to $10,000$ tokens, and for thinking models, the token limit is set to $32,000$ tokens, consistent with token limits provided by OpenRouter. We ensure that these token limits are sufficient to complete the required computations.

We use a temperature of $0.6$ and a top-p value of $0.95$ for all Gemma models. For Qwen, we use a temperature of $0.6$ and a top-p value of $0.95$ for thinking mode and a temperature of $0.7$ and a top-p value of $0.8$ for non-thinking as recommended in their documentation.[2] We find in Figure 14 that temperature does not affect the observed trends by much.

## G.5 COMPUTE DETAILS

All experiments were conducted on machines equipped with 4x NVIDIA A100 GPUs with 40/80GB memory. Frontier model evaluations were performed using OpenRouter.

---

[2]https://huggingface.co/Qwen/Qwen3-32B

```
Starting Prompt:
You are an AI assistant.  I will provide you with a dictionary and
then give you keys in groups of 2.  Your task is to keep a running
total (starting from 0) by adding the values associated with the keys
I provide.
In each turn, I'll provide 2 keys (comma-separated).  Respond with
the current running sum, enclosed in <answer> tags.
Examples:
Dictionary to maintain: 'apple': 5, 'banana': 0, 'cherry': 7,
'grape': -4, 'kiwi': 2, 'mango': -1
Example 1:  keys in groups of 2
User:  apple, banana
Assistant:  <answer>5</answer>
User:  cherry, grape
Assistant:  <answer>8</answer>
User:  kiwi, mango
Assistant:  <answer>9</answer>
Example 2:  keys in groups of 3
User:  apple, banana, cherry
Assistant:  <answer>12</answer>
User:  grape, kiwi, mango
Assistant:  <answer>9</answer>
Example 3:  keys in groups of 6
User:  apple, banana, cherry, grape, kiwi, mango
Assistant:  <answer>9</answer>
Now, here is the actual task:
Dictionary to maintain:
'doubt': -64, 'alone': 46, 'adult': 84, 'fault': -19, 'brain':
-45, 'blind': 68, ... 'coach': -31, 'alarm': 88, 'could': 25,
'cable': -32
Ready to start!
IMPORTANT: DO NOT OUTPUT ANY OTHER TEXT OUTSIDE ANSWER TAGS. Only
provide the final running sum OF ALL TURNS in <answer> tags.
```

**User:** `alarm,coach`

**Assistant:** `<answer>57</answer>`

**User:** `doubt,cable`

**Assistant:** `<answer>-39</answer>`

## H    FORMAT FOLLOWING FAILURES

In any LLM evaluation, format following failures are a common source of error that is often neglected. In our experiments, any model can have 2 types of format following failures: (1) They do not provide `<answer>` tags in their answer, and (2) They do not provide a valid integer within `<answer>` tags. To minimize format following failures, we ensure clarity in the starting prompt with clear format instructions, as well as few-shot examples. To empirically verify that model errors on our task are actually execution errors and not just format following errors in disguise, for each experiment, we also track the format failure fraction: the fraction of samples that do not correctly follow the format, with the failure being either (1) or (2). It is important to note that while we try to minimize any such error to the best of our abilities, we still count format following as a limitation of the model and hence a source of error.

Our results for format following failures are presented in Figure 16 for the experiments presented in Section 3. We observe that smaller models are more susceptible to format failures, with the Qwen3 family in particular being worse at following format instructions. Overall, the fraction for format following errors is low (around 0.1), with the Qwen3-8B being an exception. We find that the error here actually comes from the model trying to cheat, and do the entire summation *inside* the `<answer>` tags (For example, `<answer>39 + 51 = 90</answer>`). This is explicitly

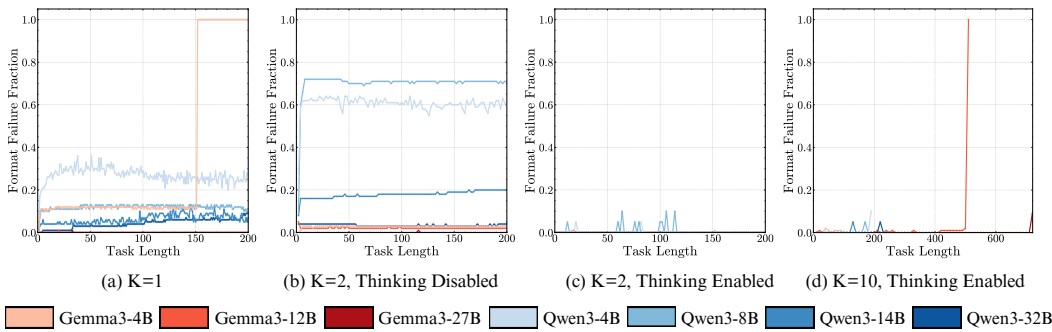

**Figure 16: Format following is not the primary mode of failure.** We analyze the fraction of errors attributed to incorrect format following for the experiments presented in Section 3. Overall, format adherence is high and not the primary source of execution errors.

forbidden as we do not allow chain-of-thought or thinking in this experiment, and thus we count this as an error. Gemma3 4B fails at later turns due to context length limitations; however, that does not affect any results, as its accuracies drop much earlier.

For the experiments presented in Figure 16, we find the Qwen3 family to be prone to format following errors in the case where we have thinking disabled for $K = 2$. We again find this to be the consequence of models trying to cheat and use extra tokens for computation inside the answer tags. This is fixed by enabling thinking. Following this, the errors in format become negligible. At $K = 10$, we see Gemma3 12B sharply rise to a format failure fraction of 1.0, again due to a full context window.

## I   CHAIN-OF-THOUGHT SELF-CONDITIONING

While our self-conditioning analysis provides clear insights for thinking models, extending this to models using Chain-of-Thought (CoT) presents some significant methodological challenges.

First, a fundamental prerequisite for reliable CoT reasoning is the inclusion of prior CoT traces in the context history. As we observed with the Gemma3 models, they often condition on the format of the context; if prior turns lack CoT traces, the models cease to generate them, even when explicitly instructed to do so. Consequently, this experiment for CoT must include the full reasoning trace for every preceding turn. This requirement immediately makes the setup practically infeasible, as the verbose nature of CoT traces would rapidly exhaust the context window limits of even frontier models.

Second, even if context length were not a constraint, the process of injecting controlled errors into CoT histories is not straightforward. A naive approach of only altering the final answer while preserving the original, correct CoT trace creates an unfaithful history. When conditioned on a history where reasoning and conclusions are contradictory, the model is no longer being tested on its execution reliability but on how it resolves inconsistency—it might learn to distrust its own reasoning, introducing a confounding variable.

The alternative is to programmatically generate flawed CoT traces. We implemented and experimented with this, and as seen in Figure 17, CoT does not mitigate the self-conditioning effect. However, this setup also introduces its own complexities. For our simple task, there are multiple distinct points of failure within a single trace: an error in the retrieval step (looking up an incorrect value) or an error in the composition step (an arithmetic mistake). A controlled experiment would need to systematically manage the type, frequency, and location of these injected errors, making the setup intractable. Even establishing a "perfectly correct" (Induced Error Rate = 0.00) baseline history is problematic. A model might have a CoT trace with flawed reasoning (e.g., a minor calculation error that cancels out), which we then replace with the correct final answer. Such a history is also unfaithful.

Given these challenges—the practical infeasibility due to context length and the difficulty of designing a faithful error injection mechanism—we limit our self-conditioning analysis to non-thinking and thinking models.

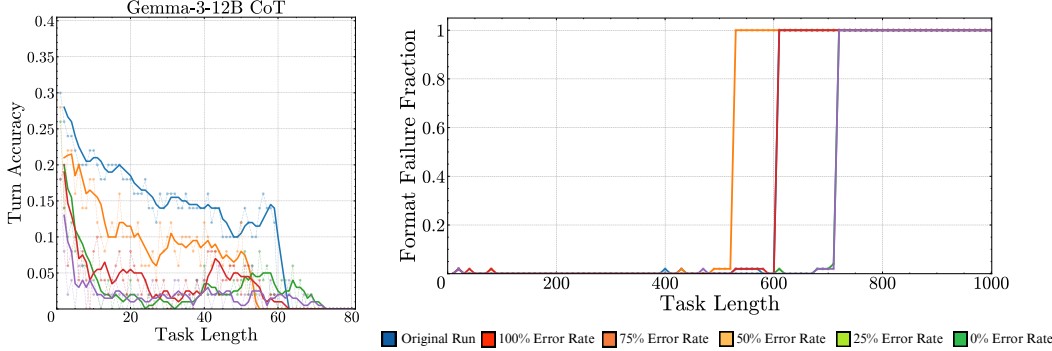

**Figure 17: CoT does not fix self-conditioning.** We observe that even with programmatically generated CoT history, the Gemma3 models cannot mitigate self-conditioning.

## J    PROOF AND ANALYSIS OF PROPOSITION 1

**Proposition 1.** *Assuming an independent and constant per-step accuracy $p$ and no self-correction, the horizon-length $H$ at which a model can achieve a success rate $s$ is given by:*

$$H_s(p) = \frac{\ln(s)}{\ln(p)}$$

*Proof.* Let $p$ be the constant probability of successfully executing a single step. Under the assumption of no self-correction, a task of length $H$ is successful only if all $H$ independent steps are executed correctly. The probability of this joint event, $P(\text{success}, H)$, is the product of the individual step probabilities:

$$P(\text{success}, H) = \underbrace{p \times p \times \cdots \times p}_{H \text{ times}} = p^H$$

This is equivalent to the Task Accuracy at turn $H$, i.e., $\text{TA}(H) = p^H$. We define the horizon-length $H$ as the number of turns at which the probability of success equals a desired rate $s$. Therefore, we set our expression for the success probability equal to $s$:

$$p^H = s$$

Solving for $H$,

$$\ln(p^H) = \ln(s) \Rightarrow H \cdot \ln(p) = \ln(s)$$

$$H_s(p) = \left\lceil \frac{\ln(s)}{\ln(p)} \right\rceil \approx \frac{\ln(s)}{\ln p}$$

This completes the proof.                                                                              □

### J.1    IMPLICATIONS FOR HORIZON LENGTH ($H_{0.5}$)

We can apply this general result to our specific metric, the Horizon Length ($H_{0.5}$), which is defined as the number of turns at which Task Accuracy drops to $s = 0.5$,

$$H_{0.5}(p) = \left\lceil \frac{\ln(0.5)}{\ln p} \right\rceil = \left\lceil -\frac{\ln(2)}{\ln p} \right\rceil$$

For analysis, we use the continuous approximation:

$$H_{0.5}(p) \approx -\frac{\ln(2)}{\ln p}$$

**Sensitivity to Small Changes in Step Accuracy.** This formulation allows us to analyze the sensitivity of the horizon length to small improvements in per-step accuracy by taking the derivative with respect to $p$,

$$\frac{dH_{0.5}}{dp} = -\ln(2) \cdot \left( -\frac{1}{(\ln p)^2} \cdot \frac{1}{p} \right) = \frac{\ln 2}{p(\ln p)^2}$$

This implies that a small change in accuracy $\Delta p$ results in a change in horizon length $\Delta H_{0.5}$ of,

$$\Delta H_{0.5} \approx \frac{\ln 2}{p(\ln p)^2} \Delta p$$

**Near-Perfect Accuracy Regime.** The effect is most dramatic when accuracy is already high. For near-perfect accuracy, let $p = 1 - \varepsilon$ where $\varepsilon \ll 1$. Using the Taylor approximation $\ln(1 - \varepsilon) \approx -\varepsilon$, we can simplify the expression for $H_{0.5}$,

$$H_{0.5} \approx -\frac{\ln(2)}{\ln(1 - \varepsilon)} \approx -\frac{\ln(2)}{-\varepsilon} = \frac{\ln 2}{\varepsilon} = \frac{\ln 2}{1 - p}$$

The sensitivity in this regime becomes,

$$\frac{dH_{0.5}}{dp} \approx \frac{\ln 2}{(1 - p)^2} \quad \Rightarrow \quad \Delta H_{0.5} \approx \frac{\ln 2}{(1 - p)^2} \Delta p$$

This demonstrates that as $p \to 1$, the improvement in horizon length for a fixed gain in step accuracy grows quadratically, highlighting the compounding benefits of scale.

