# OpenReview forum: "The Illusion of Diminishing Returns: Measuring Long Horizon Execution in LLMs"
_ICLR.cc/2026/Conference — ICLR 2026 Poster_

### Official Review · Reviewer_f3Ec · 2025-10-29

**Soundness:** 3
**Presentation:** 3
**Contribution:** 2
**Rating:** 4
**Confidence:** 3

**Summary:**

This paper investigates the scaling properties of Large Language Models (LLMs) on long-horizon tasks. The central thesis is that the perception of "diminishing returns" from scaling is an illusion created by short-task benchmarks. The authors argue that even small, diminishing gains in single-step accuracy can compound exponentially, leading to massive improvements in the total length of a task a model can successfully complete. To support this, the authors propose a distinction between "reasoning/planning" and "execution." They argue that many failures on long tasks are not failures of reasoning, but of simple execution. They introduce a synthetic, multi-step key-value dictionary addition task designed to isolate this "execution" capability by providing the model with all necessary knowledge (the dictionary) and the plan (the keys to add). The authors conclude that scaling and sequential compute both provide massive, non-diminishing benefits for the long-horizon tasks that hold true economic value.

**Strengths:**

1. The paper is well-written and the central arguments are clear to follow.
2. The synthetic experiments are thorough. The design of the "self-conditioning" experiment, which injects errors into the model's history to prove a causal link, is particularly clever and provides clear support for this specific phenomenon.

**Weaknesses:**

1.  The paper's entire set of conclusions about "long-horizon execution" is built upon an extremely synthetic and narrow task: repeatedly retrieving values from a dictionary and adding them to a running sum. The authors claim this isolates "execution," but it is highly questionable whether this type of execution is representative of the complex, diverse, and adaptive execution required in real-world agentic tasks (e.g., navigating a new website, debugging code, or managing a multi-tool workflow). Real-world execution is not typically the N-th repetition of the exact same retrieve-then-compose operation. It involves varied actions, changing states, and error correction based on new observations. Because the task is so far removed from the real-world tasks it claims to model, it is impossible to know if the findings (especially "self-conditioning") would generalize. The paper's broad conclusions about the "economic value" of LLMs are therefore built on a very tenuous foundation. The limitations section acknowledges this, but I believe it is a more fundamental flaw that challenges the paper's entire premise.

**Questions:**

1. If the problem is purely "execution," why is the solution "reasoning"?

---

> ### Author Response · Authors · 2025-11-17
> **Response 1/2**
>
> Thank you for recognizing the thoroughness of our experiments and the clarity of our central claims. Below, we once again justify the importance of controlled studies, which sometimes have to be synthetic, but we now also demonstrate how self-conditioning is a significant failure mode in realistic agentic tasks.
>
> ---
>
> ## Generalization to Real-World Agentic Tasks
>
> > Because the task is so far removed from the real-world tasks it claims to model, it is impossible to know if the findings (especially "self-conditioning") would generalize.
> >
>
> That is a fair concern. Parallel work [3] recently released annotated failed trajectories from more realistic, popular agent benchmarks like GAIA, ALFworld, and WebShop. So we now manually went through each of the annotations, and analyzed instances of self-conditioning in `Appendix A`. We estimate that ~20% of GAIA, ~48% of ALFWorld, ~33% of WebShop failures are similar to self-conditioning. Here are some such trajectory annotations as examples:
>
> - “… This set the agent on an endless loop of query reformulation and paging through irrelevant results, making success impossible …” (WebShop)
> - “… This set a precedent for all subsequent memory outputs to ignore key details about which objects had been visited what actions were performed at each and what the actual observations were …” (ALFWorld)
> - “Make a less efficient plan thats not successful, repeat the similar action thats not successful” (GAIA)
>
> The authors themselves note - *“Once an agent misremembers a fact or misjudges its progress, subsequent planning becomes systematically distorted, leading to repeated cycles of flawed action selection”.*
>
> As we highlight, there are fundamental conceptual reasons why self-conditioning occurs in today’s language models. Language model pre-training occurs under the next token prediction objective, which trains the model to output the most likely completion of its context. Once a model sees (its own) errors accumulating in its context, the most likely steps that follow have a higher probability of making errors. This is exactly what `Reviewer mGG6` points out in their example relating self-conditioning to in-context learning, which we now elaborate on in `L346-L352`.
>
> There is of course a caveat in the analysis we present above: the correctness of steps is harder and more subjective to determine on these tasks for a quantitative study. This is exactly where having a controlled study like ours, even if with a synthetic task, is extremely valuable. It allows us to isolate execution failures from planning and knowledge gaps, which is difficult in more complex agentic tasks (as we motivate in  `L206-L230`).
>
> Solving our easy toy task for long-horizons is, in some ways necessary, for coherent long-horizon LLM agents, even if not sufficient. The simplicity and minimality of our setup is a feature of our study, showing long-horizon failures can show up without complicated environments that are more challenging to study. Earlier efforts like ARC-AGI [1] and Physics of LMs [2] have been extremely influential in the community to isolate capabilities in controlled synthetic tasks which we draw inspiration from. Yet, as you acknowledged, we are candid about the limitations of our approach (`L486-L491`).

---

> ### Author Response · Authors · 2025-11-17
> **Response 2/2**
>
> ## "Economic value" of LLMs
>
> > Because the task is so far removed from the real-world tasks it claims to model … The paper's broad conclusions about the "economic value" of LLMs are therefore built on a very tenuous foundation.
> >
>
> Our discussion of “potential economic implications” of long horizon execution are only based on our conceptual discussion in `Section 2.1`, and not our experiments. We briefly discussed the economic implications only to setup the motivation for why one should care about long horizon performance when discussing scaling. The major bottlenecks to scaling are economic, and we think the length of tasks a model can do is a better measure of the economic returns from a model, so it’s probably better to look at the effects of scaling on long-horizon performance rather than single-step loss.
>
> ---
>
> ## “Reasoning” vs “Thinking”
>
> > If the problem is purely "execution," why is the solution "reasoning"?
> >
>
> We would like to clarify that we do not claim “reasoning” is the solution to “execution”. We believe the confusion may stem from our Results 5/6, where we discuss how “thinking” (sequential compute) fixes self-conditioning. We are careful to always refer to “sequential test time compute”, i.e., producing intermediate tokens before the model outputs the final answer, as “thinking”. This sequential test time compute (thinking) can help with both execution (shown in results 5 and 6) and planning.
>
> For clarity we define these terms in `L121-L129` and discuss it in `L130-L137`. What the literature currently calls “reasoning” in the context of LLMs, such as “reasoning models”, implicitly consists of both creating the plan (deciding how to solve a problem or task), and executing it (performing the required computations) [1, 2]. Under this framework, execution is a subcomponent of reasoning, that benefits from sequential test time compute in LLMs (”thinking”) based on our results. A major contribution of our work is showing a path to disentangling failures of execution from failures of planning.
>
> ---
>
> ## Premise of the paper
>
> Overall, our economic motivation for studying long-horizon execution comes from our conceptual discussion of small diminishing improvements compounding the length of tasks models can complete, which is agnostic of our choice of task for this study. Our experiments use a carefully designed synthetic task to isolate execution capabilities of LLMs, by maximally reducing the role of knowledge and planning. In trying to perform a controlled study, it is true that we move away from the messy, yet important diversity in real-world agentic tasks. We thank you for your suggestion on testing if the phenomenon of self-conditioning we identify and highlight generalizes. This led us to manually annotate real world agentic failure logs and include this analysis in our updated version, which has helped improve our paper. We hope this increases your support for our work.
>
> *[1]  Arc prize 2024: Technical report. Francois Chollet, Mike Knoop, Gregory Kamradt, and Bryan Landers. arXiv preprint arXiv:2412.04604 (8 Jan, 2025)*
>
> *[2] ICML 2024 Tutorial: Physics of Language Models. Zeyuan Allen-Zhu. Project page: https://physics.allen-zhu.com/ (July, 2024)*
>
> *[3]  AgentDebug: Where LLM Agents Fail and How They can Learn From Failures. Kunlun Zhu, Zijia Liu, Bingxuan Li, Muxin Tian, Yingxuan Yang, Jiaxun Zhang, Pengrui Han, Qipeng Xie, Fuyang Cui, Weijia Zhang, Xiaoteng Ma, Xiaodong Yu, Gowtham Ramesh, Jialian Wu, Zicheng Liu, Pan Lu, James Zou, Jiaxuan You. arXiv preprint arXiv:2509.25370 (29 Sep, 2025)*

---

> > ### Comment · Reviewer_f3Ec · 2025-11-23
> >
> > Thanks for the response. I've raised my score to 6.

---

### Official Review · Reviewer_mGG6 · 2025-10-31

**Soundness:** 3
**Presentation:** 4
**Contribution:** 3
**Rating:** 6
**Confidence:** 3

**Summary:**

This paper studies whether scaling LLMs or introducing thinking continues to yield benefits, for long-horizon tasks that requires many sequential steps. They design experiments with provided knowledge and plan to isolate execution and planning capabilities.

Their finds are:

1. longer execution are more difficult
2. larger LLMs significantly prolong the execution
3. self-condition is the main reason aside pure long context
4. larger LLMs do not affect self-condition
5. thinking alleviates self-condition
6. CoT is necessary
7. GPT-5 > Claude 4 ~ Grok-4 > others

**Strengths:**

1. The paper is well written and easy to follow.

2. The idea is well motivated; isolating the long horizon from other factors makes sense.

3. The experiment setting is well designed. (1) long horizon, knowledge, and execution are decoupled. (2) long context and self-condition are decoupled.

4. Finds are valuable to the community.

**Weaknesses:**

1. Is self-conditioning a bad behavior? From my understanding, a strong LLM with in-context reasoning **should** be self-conditioned.

For example, in this case:
```
Original number: 10
Sequence: [+3, -7, +1, +2]
Step 1: 10 + 3 = 14
Step 2: 14 - 7 = 8
Step 3: 8 + 1 = 10
...
```

In this case, a better answer should be `Step 4: 10 + 2 = 13`

2. The discussion about the relationship between self-conditioning and existing self-correction methods is necessary. The self-correction method includes agentic methods (self-refine and its following works) and reasoning behavior (aha moment).

**Questions:**

See weakness.

---

> ### Author Response · Authors · 2025-11-17
> **Response**
>
> We are happy to hear you liked our experiments and that you find our findings valuable to the community.
>
> ---
>
> ## Is self-conditioning necessarily bad?
>
> > Is self-conditioning a bad behavior? From my understanding, a strong LLM with in-context reasoning **should** be self-conditioned.
> >
>
> That’s a great point. We agree that self-conditioning and in-context learning arise from similar capabilities, which makes self-conditioning somewhat fundamental. However, there is one crucial difference that distinguishes them. In-context learning often implies learning from user and environment inputs. For example, we also provide few-shot examples of *correct* behaviour in our prompt, and the model indeed has to *condition* on these examples to learn the objective of the task.  But the crucial difference is in the “self” component of *self*-conditioning. Here, the model gets influenced by its own (potentially wrong) behaviour, and can from thereon degenerate. At some level, you are right that self-conditioning is what one would expect from a model trying to predict the most likely completion given the context. However, it implies that once the context accumulates errors, it becomes more likely for errors to appear, which can be problematic for a reasoner/agent trying to complete a long horizon task. We have now added this discussion to `L346-L352`, thanks for bringing up this relation!
>
> ---
>
> ## Self-correction
>
> > discussion about the relationship between self-conditioning and existing self-correction methods … (self-refine and its following works)…
> >
>
> Thanks for this suggestion. We have now added a discussion about self-correction works in the Related Work section from `L453-L459`. Is there anything else you would like us to add there?
>
> ---
>
> We hope our response clarifies your questions and increases your support for our work!

---

> > ### Author Response · Authors · 2025-11-26
> > **Any updates after our response?**
> >
> > Hey,
> >
> > We wanted to follow up on the rebuttal we posted a week ago. We discussed self-conditioning: whether it is necessarily bad and how it relates to self-correction.
> >
> > Does this answer your questions?

---

### Official Review · Reviewer_qfhB · 2025-10-31

**Soundness:** 3
**Presentation:** 3
**Contribution:** 3
**Rating:** 8
**Confidence:** 3

**Summary:**

This paper studies the ability of LLMs to execute many-step plans or instructions in long-horizon tasks. While this consists of planning and then execution, the paper strips away the planning part via controlled tasks where the model is given detailed instructions and "knowledge", to only test the ability to execute. The paper studies three factors: (1) the impact of model strength/scaling, (2) the impact of correct executions of earlier steps in the context (called self-conditioning in the paper), and (3) the impact of thinking.
For (1), it starts with a simple formal analysis that assumes constant and independent accuracy per step, that indicates strong gains from improving accuracy per step. The paper itself later shows that the assumptions of this analysis are not always true. To obtain a controlled empirical setting, the work uses a simple lookup and prefix-sum task, where the "dictionary" is provided. Regarding (1), they find that first, even models that are very good at one lookup or addition step can perform much worse on the long-horizon task, possibly indicating problems with state tracking. The per-step accuracy diminishes with the length of the task. Second, still, better models indeed are better at long-horizon execution. For (2), the paper shows an experiment that helps distinguish between failure due to task length versus due to errors in the execution trace, by controlling the errors in the trace. All tested models are sensitive to the errors in their history. (3) For thinking models, this sensitivity appears to go away. The final experiment measures the influence of per-step complexity, and finds that without CoT or thinking, models cannot execute complex steps.

Overall I thought this paper has some interesting findings.

**Strengths:**

- The paper shows interesting observations on the brittleness of LLMs to errors in context, and the increase in per-step error in long tasks, despite being able to execute single steps well.

- The experiments use a controlled (albeit very simple task) setting with interesting setup.

- Multiple models of different sizes are tested.

- The paper is decently written.

**Weaknesses:**

- The controlled task is rather simple, and it is a chain of always the same task (the paper mentions this already, though). I understand the controlled setting helps to get a controlled study, and that is also good. Possibly relating it to a more real-world task could have been nice.

- Some things were not fully clear -- see my questions below.

- it is possible that if the model designs a plan internally, it is different from the externally provided lookup and "plan". This is not a full weakness in that the model is not fully transparent, but it could be mentioned in the text.

- In some points, the writing was not completely clear to me, e.g.:

lines 87/88: "the per-step error rate itself rises as the task progresses. This is in contrast to humans, who typically improve at executing a task with practice." The reasoning chain is usually not "practice", hence, I was not sure how this relates to humans practicing.

lines 133/34: "In fact, failures in execution have been misattributed to limitations in reasoning or planning capabilities". If the model fails, how do you know it is planning correctly but failing to execute the plan? If you do not control for the plan like in this paper.
The description of "turn accuracy" compared to "step accuracy" is a bit confusing, as both use one state update.

Sec 3.1: the appendix explains the results with the difficulty of state tracking/management, whereas the main text talks about circuits. Why is the discussion split like that?

**Questions:**

- Sec 3: how exactly is correctness measured for the cunulative sum and addition? If e.g. the model makes a mistake in step 2, the result is wrong for the remaining steps. Is this counted as an error from step 2 onwards or just for step 2? Is this also how it is done for horizon length?

- lines 260/261: where is the CoT in Figure 12?

- You find that both state tracking and self conditioning are a problem. Do these interact, and if so, how (maybe I missed that)?

- do Sec. 3.2 and 3.3 use the same task as Sec. 3.1? In Sec. 3.3, how exactly do you vary the turn complexity? A description and/or example would be helpful.


Other minor comments:
- legend of Figure 7b: both thinking and CoT color code look the same in my pdf
- For proposition 1, add the independence assumption to the proposition statement. This may not hold in practice at all times, so good to mention it.

---

> ### Author Response · Authors · 2025-11-17
> **Response 1/2**
>
> We are glad you found our observations on self-conditioning interesting and supported with controlled studies. Thanks for the helpful feedback, here’s how we addressed it in our revision:
>
> ---
>
> ## Relation to real-world tasks
>
> > I understand the controlled setting helps to get a controlled study, and that is also good. Possibly relating it to a more real-world task could have been nice.
> >
>
> We agree with this sentiment. In `Appendix A`, we now present our analysis after manually analyzing LLM agent failure annotations from AgentErrorBench [1] to find potential instances of self-conditioning. We estimate that ~20% of GAIA, ~48% of ALFWorld, ~33% of WebShop failures are similar to self-conditioning. Here are some such trajectory annotations as examples:
>
> - “… This set the agent on an endless loop of query reformulation and paging through irrelevant results, making success impossible. …” (WebShop)
> - “… This set a precedent for all subsequent memory outputs to ignore key details about which objects had been visited what actions were performed at each and what the actual observations were …” (ALFWorld)
> - “Make a less efficient plan thats not successful, repeat the similar action thats not successful” (GAIA)
>
> The authors of [1] themselves note - *“Once an agent misremembers a fact or misjudges its progress, subsequent planning becomes systematically distorted, leading to repeated cycles of flawed action selection”.*
>
> There is of course a caveat in the analysis we present above: the correctness of steps is harder and more subjective to determine on these tasks for a quantitative study. This is exactly where having a controlled study like ours, even if with a synthetic task, is extremely valuable. Note that our motivation for the dictionary-like task was (1) the fact that many reasoning/agentic tasks can be modelled as `retrieve-then-compose` as we motivate in `L203-L215` (2) isolating execution failures from planning and knowledge gaps by making the task as simple as possible.
>
> ---
>
> ## Plan divergence
>
> > it is possible that if the model designs a plan internally, it is different from the externally provided lookup and "plan". This is not a full weakness in that the model is not fully transparent, but it could be mentioned in the text.
> >
>
> This is a great point! In real-world agentic benchmarks, even if one provides the plan to the model, it could deviate from it internally in its execution while still ending up with correct outcomes. We have now mentioned this in `L491-L493`. Indeed, this is yet another reason why our controlled study is important! Here, if the model deviates from the “plan” aka the keys we provide in the input, it is likely to end up at the wrong output.

---

> ### Author Response · Authors · 2025-11-17
> **Reponse 2/2**
>
> ## Writing clarifications
>
> > …The reasoning chain is usually not "practice", hence, I was not sure how this relates to humans practicing.
> >
>
> This line was a minor note about how humans self-correct and adapt based on their experience (both reasoning and actions) to improve, whereas models seem to worsen and make more errors. One could have viewed multi-step execution in both reasoning and agentic tasks as practice if models indeed learnt continually from it (which we agree with you, they do not). If this still seems confusing or speculative, we are happy to retract this line in the final version.
>
> > …If the model fails, how do you know it is planning correctly but failing to execute the plan? If you do not control for the plan like in this paper…
> >
>
> It is indeed hard to know this, which was the motivation for our controlled study. Existing papers [1, 2] (including one we cited) already discuss how works like illusion of thinking [3] are misattributing execution failures as planning. This is because the model generates the correct plan in its reasoning trace (as verified by human inspection), but then fails to execute it.
>
> > where is the CoT in Figure 12?
> >
>
> Thanks for pointing this out, Gemma with “thinking enabled” is actually realised using CoT, as Gemma models do not have thinking variants. We have clarified this in the updated version’s figure caption.
>
> > The description of "turn accuracy" compared to "step accuracy" is a bit confusing, as both use one state update.
> >
>
> As we mentioned in `L147-L153`, by turn we mean multiple user/assistant turns in the language model prompt format, whereas steps can be a sequence of retrieve-then-compose operations within a turn.
>
> > If e.g. the model makes a mistake in step 2, the result is wrong for the remaining steps. Is this counted as an error from step 2 onwards or just for step 2? Is this also how it is done for horizon length?
> >
>
> For “Task Accuracy” (defined in `L156`) this is counted as an error for step 2 onwards, and since this is what we measure for horizon length, it is indeed how its done. For “Step Accuracy” (defined in `L148`) this is only counted as an error in step 2.
>
> > You find that both state tracking and self conditioning are a problem. Do these interact, and if so, how (maybe I missed that)?
> >
>
> Indeed, when decoupling each step into individual operations in `Figure 14 (b)` we do find that on the ablated task of independent additions in each turn, models make errors much more slowly (14 a), while when adding to a previous state, the errors appear much faster. However, note that in `Figure 14` the Y axis is the “Task accuracy”, which counts an error when any mistake was made in the entire prefix. On the other hand, our self-conditioning results are based on “step accuracy”, where only the error of execution of the current step is measured, and worsens over time. While self-conditioning can apply to worsen model performance on any task, state tracking mistakes are specific to our setup. They may interact, in the sense that state tracking can worsen over time, per step, due to self-conditioning, though we did not investigate this in detail in our study.
>
> > do Sec. 3.2 and 3.3 use the same task as Sec. 3.1? In Sec. 3.3, how exactly do you vary the turn complexity? A description and/or example would be helpful.
> >
>
> Yes, `Section 3.2` and `Section 3.3` use the same task as `Section 3.1`. In `Section 3.3`, we measure the maximum number of steps (complexity) a model can execute per turn (turn complexity). To be precise, instead of a multi turn conversation like previous sections, the turn complexity is realized as the number of keys we prompt the model to process in a single turn.
>
> > legend of Figure 7b: both thinking and CoT color code look the same in my pdf. For proposition 1, add the independence assumption to the proposition statement. This may not hold in practice at all times, so good to mention it.
> >
>
> We had marked the CoT model bars with black stripes, we have changed it to white stripes now for better readability. We have now also added the Independence assumption to the proposition statement.
>
> ---
>
> Thank you for helping us improve our paper!
>
> *[1]  A comment on "the illusion of thinking": Reframing the reasoning cliff as an agentic gap. Sheraz Khan, Subha Madhavan, and Kannan Natarajan. arXiv preprint arXiv:2506.18957 (23 June, 2025)*
>
> *[2] Solving a Million-Step LLM Task with Zero Errors. Elliot Meyerson, Giuseppe Paolo, Roberto Dailey, Hormoz Shahrzad, Olivier Francon, Conor F. Hayes, Xin Qiu, Babak Hodjat, Risto Miikkulainen. arXiv preprint arXiv:2511.09030 (12 Nov, 2025)*
>
> *[3] The illusion of thinking: Understanding the strengths and limitations of reasoning models via the lens of problem complexity. Parshin Shojaee, Iman Mirzadeh, Keivan Alizadeh, Maxwell Horton, Samy Bengio, and Mehrdad Farajtabar. arXiv preprint arXiv:2506.06941 (18 Jul, 2025)*

---

> > ### Author Response · Authors · 2025-11-26
> > **Any updates after our response?**
> >
> > Hey,
> >
> > We wanted to follow up on the rebuttal we posted a week ago. We incorporated your suggestions on testing our observations on a real-world task, and improved the writing based on your feedback.
> >
> > Does this address your suggestions?

---

### Official Review · Reviewer_42Ay · 2025-11-03

**Soundness:** 3
**Presentation:** 4
**Contribution:** 3
**Rating:** 6
**Confidence:** 4

**Summary:**

This paper proposes the idea that the common notion of "diminishing returns" on single-step tasks (as models are scaled up) should not be taken to mean that there is no value in pushing for those small increases, as these seemingly small increases can lead to an exponentially large benefit on long-range execution. In particular, the authors propose a synthetic tasks (of essentially computing "prefix sums" of a sequence of numbers revealed via a key-value mechanism) as well as setups/metrics to disentangle execution ability (the focus of this study) from the ability to properly plan and/or retrieve knowledge. With detailed studies on this benchmark, the authors provide evidence supporting their claims about diminishing returns being an illusion, LLM trends for long-horizon execution tasks, etc.

**Strengths:**

* The question raised by the authors, namely whether the diminishing rate of improvement in single-step tasks is actually a real problem or not, is an interesting and practically valuable one. The authors' proposal of considering the max task execution length as a key metric is also of importance.

* The study is clearly motivated and the empirical section is well-organized with clearly labeled setup and results section.

* The authors raise a fairly large number of interesting questions and design specific experiments to test them.

* The synthetic long-horizon execution task seems well-suited to the authors' desire to isolate away effects of knowledge retrieval and planning. The idea of treating planning and retrieval as "key value" problems is unique, and it allows the authors to "provide" the plan and knowledge at each step as appropriate.

* Some of the results are quite surprising and creatively supported by the authors with explanation hypotheses. E.g., that single-step accuracy decreases with task length seems very unintuitive at first sight, but then the authors try out two hypotheses and find their combination to be a useful explanation of this phenomenon.

**Weaknesses:**

* The synthetic task proposed here boils down to retrieving and summing up (all prefix subsequences of) $t \times K$ numbers, where each number has 2 digits. While this task fits the authors' desire to isolate out planning and retrieval issues, it's **not exactly a sequential task** as referred to by the authors when discussing some experiments. This is important as, while sequential tasks are known to benefit from "reasoning" tokens or chain-of-thought, this is not really the case for adding $N$ numbers. Formally, this problem is well-known to lie in the complexity class $TC^0$ of *fully parallelizable* problems! This is a bit counter-intuitive, but happens to be true --- that one can add $N$ $N$-bit numbers completely in parallel in just a few steps. In fact, when a Transformer-based LLM compute attention over $N$ previous tokens, this sum is precisely what it computes. Thus, in a way, the synthetic task here isn't quite the perfect fit for what the authors really need. (An alternative task would be where 2-digit numbers are replaced with $5 \times 5$ size *matrices*, and instead of sum, the task is to *multiply* the matrices; this is *not* known to be fully parallelizable, making it a better fit for some of the authors' narrative around CoT and reasoning tokens being helpful.)

* That said, the experiments even with the proposed "prefix sum" task do seem to support the authors' claims in practice. I do find some details a little hard to believe/understand, though. For instance, Fig 7(a) seems to suggest that without CoT, strong models like Deepseek-V3 an Kimi-k2 can barely add 4-6 numbers of 2 digits each. I would have expected them to be able to handle longer sequences.

* The discussion in section 2.1 (diminishing returns) feels not very convincing or deep. For one, Proposition 1 seems to follow directly from the observation that if each step *independently* succeeds (a strong assumption) with probability $p$, then $T$ steps will succeed with probability $s = p^T$. If this is the reasoning, it would be better to spell it out in the main text, as readers will readily connect with this common understanding of single-step errors compounding in multi-step scenarios.

* Similarly, it is well-understood that single-step errors compound in multi-step uses. It would be helpful for the authors to discuss this. The common "use" of this observation is that a small *decrease* in single-step accuracy often leads to a large *drop* in multi-step accuracy. What the authors are saying is really the same statement, looked from the other direction: namely, a small *increase* in single-step accuracy often leads to a large *increase* in multi-step accuracy. These kinds of connections both help understand where the proposed observation is coming from and why it's reasonable. Yet, the discussion is not framed this way.

* In the same section, in Fig 2, the claim that a small increase in step accuracy (x-axis) leads to a large increase in horizon length (y-axis) appears to really be only true when the slope of the curve is significantly larger than 1, which happens when step accuracy is already in the range of 90% to 100%. Note that for long-range tasks to be meaningfully solved, $H_{0.75}$ or $H_{0.9}$ are much more realistic measures than $H_{0.5}$. In other words, the exponential increase observation (with small increases in step accuracy) seem to "work out" only when single-step accuracy is already very high, in the 90%+ range. This seems important but is not discussed.

**Questions:**

Could the authors comment on the points raised in the weaknesses section and how they might address it? I think the paper is thought-provoking and I am open to increasing my score contingent on the authors' willingness to address some of the points raised.

---

> ### Author Response · Authors · 2025-11-17
> **Response**
>
> Thank you for the insightful comments and suggestions, we found them to be quite thought provoking!
>
> ---
>
> ## Sequentiality of the task
>
> > While this task fits the authors' desire to isolate out planning and retrieval issues, it's not exactly a sequential task as referred to by the authors when discussing some experiments. This is important as, while sequential tasks are known to benefit from "reasoning" tokens or chain-of-thought, this is not really the case for adding numbers.
> >
>
> Thanks for pointing out the TC0 connection to us. This is quite helpful. We would like to begin by noting: since most of our experiments are in the multi-turn setting, where the keys to retrieve and add only appear sequentially, the model cannot parallelize the task.
>
> However, it is true that in the single turn long horizon execution benchmarking we did in Figure 7, the task is parallelizable in theory. Although in practice, it seems like frontier language models do not learn to solve this task in parallel, and only solve the task when computing it sequentially with chain of thought. However, we agree that the prefix sum task may not be future-proof to test the number of sequential operations models can perform (given that we agree that better models could solve this task by parallelizing it internally), and now acknowledge this in Limitations `L496-L499`.
>
> To address this, we now repeat the experiment in Figure 7 with the chained matrix ($2 \times 2$) multiplications as you had suggested in `Appendix B`. As shown in `Figure 8`, even frontier non-thinking models like DeepSeek V3 and Kimi K2 fail to multiply even 2 matrices whereas their thinking counterparts DeepSeek R1 can do 12 and Kimi K2 Thinking can perform 14 correctly. This reaffirms our finding of CoT and thinking tokens being helpful.
>
> ---
>
> ## Performance of non-thinking models
>
> > I do find some details a little hard to believe/understand, though. For instance, Fig 7(a) seems to suggest that without CoT, strong models like Deepseek-V3 an Kimi-K2 can barely add 4-6 numbers of 2 digits each. I would have expected them to be able to handle longer sequences.
> >
>
> Thats a great observation, we found them quite interesting too! Here are anonymous links to conversations with our prompt showing that DeepSeek V3 (https://chat.deepseek.com/share/ptnlwrb4z2sxe38zfc) and GPT 5 Instant (https://chatgpt.com/share/6918ac8c-4bec-800f-a0fc-466614344836), without CoT, fail to add the values of even 4 keys at a time. They are wrong from the very first turn, where the expected answer is -39.
>
> While the theoretical results do suggest that transformers can perform this TC0 operation, in practice, it seems like they cannot, whereas their thinking variants can. We can see that DeepSeek R1 (https://chat.deepseek.com/share/hs7r0f9r1mk9ztb2l4), GPT 5 Thinking (https://chatgpt.com/share/6918d2e7-3b6c-8001-9711-e9b586f02b44) and Gemini 2.5 Flash thinking (https://gemini.google.com/share/68da83b935cc) do give the correct answers throughout the entire conversation.
>
> ---
>
> ## Fixes to the paper
>
> > …spell it out in the main text, as readers will readily connect with this common understanding of single-step errors compounding in multi-step scenarios...
> >
>
> This is indeed the core intuition that is used to derive the relation between horizon length for a given desirable given success rate. We have now stated this on `L171`.
>
> > Note that for long-range tasks to be meaningfully solved, $H_{0.75}$ or $H_{0.90}$ are much more realistic measures than $H_{0.50}$
> >
>
> We agree, and this was our motivation to present various success rates in `Figure 2`. We only use $H_{0.50}$ for our discussion in `Section 2.1` to draw a parallel to the METR study that uses the same threshold. In our actual experiments (`Figure 7`), we use a higher, more realistic success rate of 80%, as mentioned in `L411`.
>
> > Similarly, it is well-understood that single-step errors compound in multi-step uses … These kinds of connections both help understand where the proposed observation is coming from and why it's reasonable…
> >
>
> Thanks for pointing this out, we have now added a discussion on this in `L172-L174`.
>
> > In other words, the exponential increase observation (with small increases in step accuracy) seem to "work out" only when single-step accuracy is already very high, in the 90%-range. This seems important but is not discussed.
> >
>
> We agree, and tried to be clear about this point by stating “high-accuracy regime” in the figure caption and surrounding discussion. We have further clarified this in the updated version in `L173`.
>
> ---
>
> We hope these clarify all your concerns and we are happy to discuss further. Your review has greatly helped improve our work, thanks a lot!

---

> > ### Author Response · Authors · 2025-11-26
> > **Any updates after our response?**
> >
> > Hey,
> >
> > We wanted to follow up on the rebuttal we posted a week ago. We incorporated your suggestions on sequentiality and fixes to the writing, and showed more evidence of how models fail without CoT on our current setup.
> >
> > Does this address your suggestions?

---

### Official Review · Reviewer_xdtj · 2025-11-11

**Soundness:** 2
**Presentation:** 3
**Contribution:** 2
**Rating:** 6
**Confidence:** 3

**Summary:**

This paper investigates whether the apparent diminishing returns from scaling LLMs are an illusion caused by evaluating them on short-horizon tasks. It identifies execution errors instead of reasoning failures as the main limitation in long-horizon task performance. The authors design a controlled synthetic setup that isolates execution by providing explicit plans and knowledge, enabling the measurement of how many steps a model can reliably execute. They discover that even marginal single-step accuracy improvements compound exponentially into longer horizon lengths, introduce the notion of self-conditioning and show that thinking models mitigate this effect. Experiments across models show non-diminishing gains from scaling.

**Strengths:**

* The paper presents a well-structured experimental design using a key–value retrieval task.
* The paper uncovers a new degradation source of models compounding their own mistakes, which is novel.
* The authors evaluate multiple model families, showing consistent scaling trends.

**Weaknesses:**

* The arithmetic-like setup may not capture the diverse error dynamics in natural environments. Extending to real-world planning or tool-use tasks could strengthen claims of generality.
* The focus is primarily on model size and sequential test-time compute, without exploring complementary factors such as training curricula, architecture modifications, or memory-augmented mechanisms that may interact with long-horizon execution.

**Questions:**

See suggestions in weaknesses.

---

> ### Author Response · Authors · 2025-11-17
> **Response**
>
> We are glad you found our experimental design well-structured and liked the new degradation source we found (self-conditioning). We hope the response below addresses your concerns.
>
> ---
>
> ## Extending to Real-World Tasks
>
> > The arithmetic-like setup may not capture the diverse error dynamics in natural environments. Extending to real-world planning or tool-use tasks could strengthen claims of generality.
> >
>
> We designed the synthetic task for a controlled study of model execution capabilities. We agree with your suggestion, so we decided to see if self-conditioning, one of our main contributions, also holds in more realistic agentic tasks, with our analysis now added as `Appendix A.` We manually classify LLM agent failure annotations from AgentErrorBench [1] to find instances of self-conditioning. We estimate that ~20% of GAIA, ~48% of ALFWorld, ~33% of WebShop failures are similar to self-conditioning. Here are some such trajectory annotations as examples:
>
> - “… This set the agent on an endless loop of query reformulation and paging through irrelevant results, making success impossible. …” (WebShop)
> - “… This set a precedent for all subsequent memory outputs to ignore key details about which objects had been visited what actions were performed at each and what the actual observations were …” (ALFWorld)
> - “Make a less efficient plan thats not successful, repeat the similar action thats not successful” (GAIA)
>
> The authors of [1] themselves note - *“Once an agent misremembers a fact or misjudges its progress, subsequent planning becomes systematically distorted, leading to repeated cycles of flawed action selection”.*
>
> There is a caveat in the analysis we present above: the correctness of steps is harder and more subjective to determine on these tasks for a quantitative study. This is exactly where having a controlled study like ours, even if with a synthetic task, is extremely valuable. It allows us to isolate execution failures from planning and knowledge gaps, which is difficult in more complex agentic tasks (as we motivate in  `L206-L233`).
>
> ---
>
> ## Effect of architecture
>
> > The focus is primarily on model size and sequential test-time compute, without exploring complementary factors such as training curricula, architecture modifications, or memory-augmented mechanisms that may interact with long-horizon execution.
> >
>
> That is a great point! While the main focus of our work is on isolating long horizon execution and studying the effects of scaling, we agree that memory mechanisms and architecture modifications can extend execution horizon lengths.
>
> We now test Qwen3-Next 80B-3B active with CoT in the same setup as `Figure 7 (b)`, which combines linear attention (Gated DeltaNet) with standard attention, explicitly motivated for long-context performance. We find a 80% execution horizon of 352 steps, which is much better than 80 steps for the larger, standard attention only, Qwen3 235B-22B active with CoT. The new results are added to the updated `Figure 7`.
>
> This is evidence of architectural modifications improving long horizon execution, although more controlled ablations might be needed to isolate the causal effect of the architecture change. We think this shows the value of our the task we design, as a controlled testbed for future work. Moreover, we are excited about new work that does indeed show the benefits of training curricula [2] and benefits of breaking down long tasks into subcomponents [3].
>
> ---
>
> We hope these clarifications increase your support for our work.
>
> *[1] AgentDebug: Where LLM Agents Fail and How They can Learn From Failures. Kunlun Zhu, Zijia Liu, Bingxuan Li, Muxin Tian, Yingxuan Yang, Jiaxun Zhang, Pengrui Han, Qipeng Xie, Fuyang Cui, Weijia Zhang, Xiaoteng Ma, Xiaodong Yu, Gowtham Ramesh, Jialian Wu, Zicheng Liu, Pan Lu, James Zou, Jiaxuan You. arXiv preprint arXiv:2509.25370 (29 Sep, 2025)*
>
> *[2] h1: Bootstrapping LLMs to Reason over Longer Horizons via Reinforcement Learning.
> Sumeet Ramesh Motwani, Alesia Ivanova, Ziyang Cai, Philip Torr, Riashat Islam, Shital Shah, Christian Schroeder de Witt, Charles London. arXiv preprint arXiv:2510.07312 (15 Oct, 2025)*
>
> *[3] Solving a Million-Step LLM Task with Zero Errors. Elliot Meyerson, Giuseppe Paolo, Roberto Dailey, Hormoz Shahrzad, Olivier Francon, Conor F. Hayes, Xin Qiu, Babak Hodjat, Risto Miikkulainen. arXiv preprint arXiv:2511.09030 (12 Nov, 2025)*

---

> > ### Author Response · Authors · 2025-11-26
> > **Any updates since our response?**
> >
> > Hey,
> >
> > We wanted to follow up on the rebuttal we posted a week ago. We showed the generality of self-conditioning to real-world agentic tasks, and also showed potential benefits of changing the architecture by comparing the new hybrid attention Qwen3 model.
> >
> > Does this address your suggestions?

---

### Author Response · Authors · 2025-12-03
**Final Author Remarks**

We thank the reviewers for their constructive feedback which helped further improve our paper during the rebuttal phase.

We appreciate the recognition of our experimental design being sound, novel, and of practical value. `Reviewer xdtj` praised the *"well-structured experimental design"* and noted that the identification of self-conditioning *"uncovers a new degradation source".*`Reviewer 42Ay` highlighted that we *“raise a fairly large number of interesting questions”* and *“the results are quite surprising and creatively supported”* by using a *“unique”* experiment design that is *“well-suited”* to our goal: *“isolate (execution, by taking) away the effect of planning and knowledge”*. `Reviewer qfhB` highlighted the *"interesting observations on the brittleness of LLMs to errors in context",* while `Reviewer f3Ec` described the self-conditioning experiments as *"thorough"* and *"particularly clever".* `Reviewer mGG6` found the insights *"valuable to the community"* and reviewers also the found the paper to be *“well written”* (`Reviewers [qfh, mGG6, f3Ec]`), with *“well-organized”* experiments (`Reviewer 42Ay`).

---

We now summarize the improvements we made during the rebuttal phase to address all weaknesses brought up by the reviewers.

| Suggestion | Improvements |
| --- | --- |
| Applicability to real-world tasks (`Reviewers [xdtj, qfhB, f3Ec]` ) | We add a new analysis, finding potential instances of self-conditioning in LLM agent failure annotations released in AgentErrorBench. We estimate that ~20% of GAIA, ~48% of ALFWorld, ~33% of WebShop failures reflect self-conditioning, confirming that the findings of our controlled study are important across popular agentic benchmarks. **Following this,** `Reviewer f3Ec` **increased their score from 4 to 6.** |
| Sequentiality of the task (`Reviewer 42Ay`) | As suggested by the reviewer, we repeated the experiment in Figure 7 with the chained matrix multiplications, which is an inherently sequential task even in the single-turn setting, adding results to `Appendix B`. We observe consistent observations in `Figure 8`, with frontier non-thinking models failing to multiply even 2 matrices, while their thinking counterparts can do 12-14 correctly. We also provide anonymous links to conversations with DeepSeek v3 and GPT 5 Instant, showing that without CoT, these models indeed fail to add the values of even 4 keys at a time, which the reviewer had found “surprising”. This reaffirms the importance of sequential test time compute for single-turn long executions. `Reviewer 42Ay` could not respond, but stated in their review they would be **willing to raise their score (originally 6)**. |
| Effect of Architecture (`Reviewer xdtj`) | To understand the effects of new architectures, we now tested Qwen3-Next 80B-3B active with CoT in the same setup as `Figure 7 (b)`, which combines linear attention (Gated DeltaNet) with standard attention. We find a 80% success rate execution horizon of 352 steps, which is much better than 80 steps for the larger, standard attention only, Qwen3 235B-22B active with CoT.  |
| Relation between self-conditioning, in-context learning, self-correction (`Reviewer mGG6`) | We clarified that while in-context learning from environment or user input is desirable, in self-conditioning, the model conditions on its own potentially wrong generations instead of ground truth examples, which can be undesirable. We added a discussion of self-correction, and how improving it could mitigate self-conditioning in our Related Work section.  |

We also incorporated all suggestions about wording, with all changes we made to the PDF colored in $\color{blue}{blue}$.

Overall, 4/5 reviewers did not get time to acknowledge our response, so we hope this short summary helps the new AC understand how we addressed all reviewer feedback.

---

### Meta-Review · Area_Chair_VEhn · 2026-01-04

**Summary:**

This paper investigates the execution capabilities of LLMs on long-horizon tasks, proposing that marginal improvements in single-step accuracy lead to exponential gains in total task completion length. The primary concerns raised by the reviewers centered on the ecological validity of the synthetic prefix-sum task and its ability to generalize to complex, real-world agentic failures. Reviewers also questioned the theoretical sequentiality of the task, noting that multi-digit addition is formally parallelizable within the complexity class, and sought clarification on whether "self-conditioning" was a negative failure mode or merely a byproduct of desirable in-context learning. Furthermore, Reviewer 42Ay observed that the claimed exponential benefits of scaling single-step accuracy primarily manifest in high-accuracy regimes exceeding 90%. In response, the authors significantly strengthened the submission by introducing an analysis of real-world benchmarks—GAIA, ALFWorld, and WebShop—which attributed 20-48% of trajectories to self-conditioning failure modes. They also added a truly sequential chained matrix multiplication task that confirmed the necessity of sequential test-time compute for long horizons. These revisions, alongside clear distinctions between error propagation and ground-truth learning, successfully resolved the initial skepticism, leading to a consensus that the paper provides a novel and well-structured framework for isolating execution failures from planning.

**Reviewer Concerns:**

The reviewers' primary concerns regarding the synthetic nature of the task and its generalization to real-world scenarios were effectively addressed during the rebuttal through an analysis of the AgentErrorBench dataset, which revealed that self-conditioning accounts for approximately 20% to 48% of failures in benchmarks like GAIA, ALFWorld, and WebShop. Additionally, the technical concern raised by Reviewer 42Ay regarding the theoretical parallelizability of the prefix-sum task was mitigated by the inclusion of a truly sequential chained matrix multiplication task, which confirmed the necessity of sequential test-time compute. The authors also successfully clarified the distinction between self-conditioning and in-context learning, noting that the former leads to an undesirable "death spiral" of error accumulation. However, some concerns remain partially outstanding, specifically the observation that the exponential benefits of scaling single-step accuracy are primarily constrained to models already in the high-accuracy regime above 90%. Furthermore, while the authors provided strong qualitative evidence, the quantitative measurement of execution failures in complex, natural environments remains inherently more subjective than in controlled synthetic settings. Finally, the speculative parallel drawn between model degradation and human practice remains an interesting but unproven observation that the authors expressed willingness to retract to maintain focus.

**Reviewer Scores:**

The reviewers' scores likely would have seen a net positive shift had they been able to fully engage in the post-rebuttal discussion, as evidenced by Reviewer f3Ec already increasing their score from 4 to 6 after reviewing the authors' analysis of real-world agent failure logs. Reviewer 42Ay explicitly stated a willingness to raise their initial score of 6 contingent on the authors addressing concerns regarding task sequentiality; given the successful addition of the chained matrix multiplication task, which confirmed the necessity of thinking tokens for truly sequential operations, it is highly probable their score would have moved toward a stronger acceptance. Reviewers xdtj and mGG6, who both started with a 6, likely would have maintained or marginally increased their ratings as the authors provided the requested architectural ablations for Qwen3 and clarified the fundamental distinction between self-conditioning and beneficial in-context learning. While Reviewer qfhB did not participate in the final discussion, their initial rating of 8 already represented a strong acceptance, and the authors' thorough resolution of their requests for writing clarifications and the addition of the independence assumption to Proposition 1 would have likely solidified this positive assessment. Overall, the authors successfully addressed the specific technical triggers each reviewer identified as barriers to higher scores, suggesting that full participation would have resulted in a clearer consensus toward acceptance.

---

### Decision · Program_Chairs · 2026-01-26

Accept (Poster)